# Flexible metal-organic framework films for reversible low-pressure carbon capture and release

Sumea Klokic [1] ✉, Benedetta Marmiroli [2], Giovanni Birarda [3], Florian Lackner [4], Paul Holzer[2], Barbara Sartori [2], Behnaz Abbasgholi-NA[5], Simone Dal Zilio [5], Rupert Kargl [4], Karin Stana Kleinschek [4], Chiaramaria Stani [1], Lisa Vaccari [3] & Heinz Amenitsch [2] ✉

Transitioning metal-organic frameworks (MOFs) from laboratory-scale to carbon dioxide ($CO_2$) capture and storage applications (CCS) requires in-depth understanding of their adsorption properties and structural stability, especially for film assemblies. However, evaluating their performance is challenging, particularly under low or moderate $CO_2$ pressure conditions, which are key for cost and performance efficiency. Herein, we explore the low-pressure $CO_2$ uptake and release within flexible Zn-based MOF film structures with diverse ligand functionalities, employing quartz crystal microbalance, synchrotron radiation-based infrared spectromicroscopy and grazing incidence wide-angle X-ray scattering measurements. To investigate $CO_2$ adsorption and its interaction with Zn-MOF pores, we exploited the framework's flexibility by triggering structural changes and thus variations of the pore-environment using two stimuli, temperature and light. Results show considerable promise for stimuli-induced on-demand $CO_2$ capture and release at low pressures, demonstrating structural reversibility under near-ambient conditions and highlighting the potential of tailored MOF film structures in advancing green CCS-technologies.

Advancing the integration of highly porous and adaptable MOFs – structures composed of metal nodes interconnected by organic linker molecules – into practical applications requires improvements in their industrial scalability, especially in carbon dioxide capture and storage technologies (CCS)[1–5]. This involves configuring them into large-scale CCS-compatible assemblies[6,7], such as membrane or thin-film structures[8–10], while maintaining their sorption properties and structural stability[11,12]. The pressing need to develop green CCS technologies is evident given that fossil fuels encompass over 85% of the global energy production, where $CO_2$ is a significant byproduct that accounts up to 60% in global warming[13,14]. Here, MOFs offer great promise due to their high $CO_2$ uptake capacity[15] and because known drawbacks of current CCS technologies, such as corrosion and high energy costs in the case of amine scrubbing[13], or energy-intensive adsorbent regeneration as for zeolites or molecular sieves that typically result in additional $CO_2$ emissions[16–19], could be overcome[20]. Owed to the chemical versatility of MOF structures, their performance regarding their $CO_2$ capacity and binding sites has been studied to great extent, particularly in the context of gas mixtures, direct air or high-pressure $CO_2$ capture[1–3,21–23].

However, only a limited number of studies have been reported for MOFs regarding their $CO_2$ capture and release under moderate or

[1]CERIC-ERIC, Trieste, Italy. [2]Institute of Inorganic Chemistry, Graz University of Technology, Graz, Austria. [3]Elettra Sincrotrone Trieste, Trieste, Italy. [4]Institute of Chemistry and Technology of Bio-Based Systems, Graz University of Technology, Graz, Austria. [5]IOM-CNR, Laboratorio TASC, Trieste, Italy. ✉e-mail: sumea.klokic@elettra.eu; amenitsch@tugraz.at

low-pressure conditions[6,14,16], despite the clear evidence that minimal gas adsorption is crucial for cost reduction while maintaining good process performance, as highlighted by Petit and co-workers[24]. For low-pressure $CO_2$ capture, MOF systems that meet the previously outlined demand of CCS-compatible assemblies, showing moreover high stability and simple regeneration for long-term usage, are required[23]. Exemplarily, this can be achieved when fabricating MOFs as film structures. Enhancing the system's performance and selectivity relevant for CCS imposes the understanding of the physicochemical and structural performance of MOF films towards low-pressure $CO_2$. Such key parameters can only be deduced through suitable experimental approaches, and only then, one can enhance the system's performance and selectivity relevant for CCS[11,14,24].

Although various methods have been reported, including sorption isotherms, powder or single crystal X-ray diffraction[25], vibrational spectroscopies[12,26,27] and $^{13}C$-NMR[28], only a subset is directly applicable to MOF thin-film assemblies under low $CO_2$ pressures. For instance, Knebel et al.[29] demonstrated $CO_2$ permeation in an ultrathin UiO-67 membrane using a Wicke-Kallenbach cell, but this approach is limited to permeable surfaces and is not suitable for solid substrates. Other infrared spectroscopic studies successfully demonstrated gas uptake by oriented MOF film structures, yet not for $CO_2$[30]. It is relevant to mention that attenuated total reflectance (ATR) infrared spectroscopy has proven promising for porous ZIF-8/ionic liquids[31] or oriented films grown directly onto the ATR crystal[32]. Nevertheless, this sampling approach has some limitations: first, not every MOF film fabrication approach allows the direct grafting of the structure onto the crystal and second, the limited IR evanescent field penetration becomes troublesome and hampers characterization of the film's performance under $CO_2$ load for thicker films as typically encountered in multi-layered assemblies.

Hence, in this study, three approaches are presented to investigate the reversible $CO_2$ uptake and release by MOF films, which can be readily applied to investigate structures of any topology and irrespective of the film fabrication protocol with the possibility to use various substrate types. Three *operando* methods have been exploited to this aim, namely Grazing Incidence Wide Angle X-Ray Scattering

(GIWAXS), Fourier Transformed Infrared (FT-IR) spectromicroscopy and Quartz-Crystal Microbalance with Dissipation monitoring (QCM-D; Fig. 1). This combination ensures to successfully unravel chemical features along with structural changes within the films, while QCM-D measurements allow to quantify the amount of adsorbed $CO_2$. Especially for oriented MOF assemblies, features related to the uptake of guest molecules are often subtle and to resolve such, synchrotron radiation is essential. Exemplarily, Fischer and co-workers[11] successfully used Grazing Incidence synchrotron X-Ray Diffraction (GIXRD) to resolve structural features of liquid-phase epitaxial MOF films under methanol vapour pressure, while GIWAXS measurements employing synchrotron radiation have proven effective to track changes in multi-layered or epitaxial MOF films by some authors involved in this study[33,34].

Herein, we examined the interaction of $CO_2$ with functionalized MOF film structures and the reversibility of $CO_2$ ad-/desorption under moderate gas-flow conditions. To this aim, heteroepitaxially grown Zn-based MOF films of the type $Zn_2L_2DABCO$ grown on the epitaxial $Cu_2BDC_2$-on-$Cu(OH)_2$ substructure[35] were assessed by varying the bulkiness and electron-donating properties of bridging ligands $L_2$ (see Fig. 2)[36]. The selection of L was inspired by reported examples that demonstrated i.e., enhancements in $CO_2$ adsorption for amino-groups in bulk UiO-66[37], improved stability towards moisture when introducing hydrophobic groups such as methyl[38], or an increased structural flexibility for alkyl ether groups in Zn-based MOFs[39,40]. This characteristic was utilized to investigate the adsorption of $CO_2$ molecules within the $Zn_2L_2DABCO$ pores, focusing on how stimuli-induced pore size transitions (large-pore (LP) to narrow-pore phase (NP), vide infra)[41] and the resulting structural and chemical environment variations influence the adsorbed $CO_2$ molecules. The two external stimuli used in this work were temperature and light, where the latter shows particular promise for facilitating on-demand $CO_2$ uptake and remotely controlled release, due to its ease of application and ubiquity. To test responsiveness to light within the $Zn_2L_2DABCO$ films, the photo-active azobenzene molecule was introduced into the functionalized pores of the most flexible Zn-MOF structures[29,42,43]. Our findings demonstrate that light induces a significant structural response in

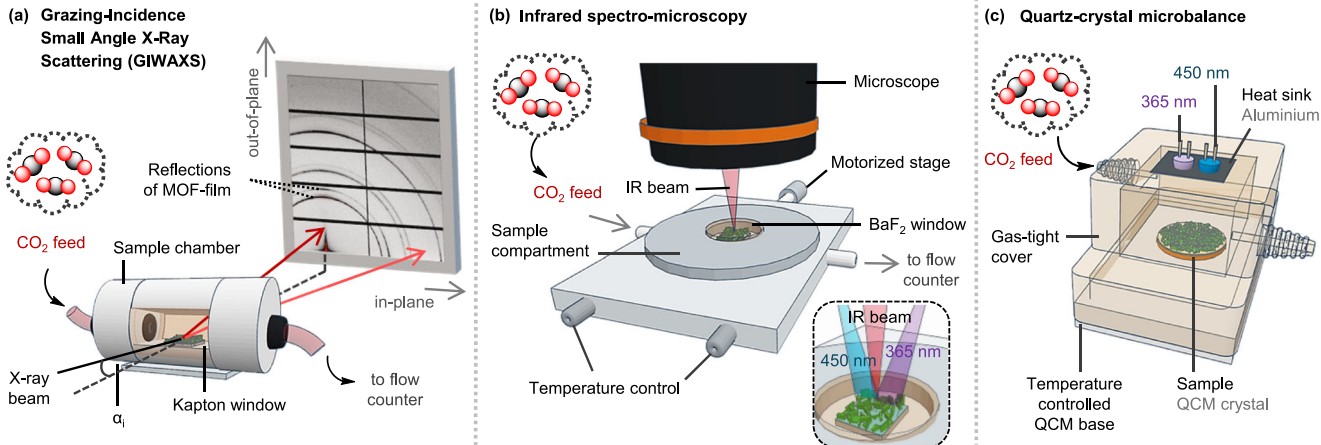

**Fig. 1 | Device schematics to track low-pressure $CO_2$ uptake and release.**
**a** Outline of the Grazing-Incidence Wide Angle X-ray scattering (GIWAXS) set-up for the investigation of the crystallographic features during $CO_2$ uptake by the Zn-MOF films. The $CO_2$ feed gas was brought into a sample chamber equipped with Kapton windows for the incident (black arrows, incident angle = $\alpha_i$) and scattered X-ray beam (red arrows). Diffraction patterns of the respective MOF films comprising characteristic reflections of the MOF films were monitored along the in-plane and out-of-plane direction (gray arrows). The $CO_2$ gas was led into an exhaust water-container acting as a flow counter. **b** IR spectromicroscopy set-up during $CO_2$ uptake by the MOF films. The sample was placed in a temperature-controlled

compartment with a motorized stage, equipped with gas transport connectors, top- and bottom-sealed with a $BaF_2$ window transparent for wavelengths up to the UV-range (see inset). **c** Quartz-crystal microbalance with dissipation (QCM-D) with a self-build, proprietary measurement cell set-up for quantifying the $CO_2$ uptake by the MOF films equipped with LED diodes for photo-switching experiments (365 nm (purple), 450 nm (blue)), which were soldered onto an aluminium heat sink. The sample was directly grown onto the QCM-D crystal and placed inside the gas-tight cell compartment equipped with a temperature-controlled base. Portions of the schematics, including elements of the GIWAXS and IR microscopy setups, are adapted from work created in Tinkercad licensed under CC BY-SA 3.0.

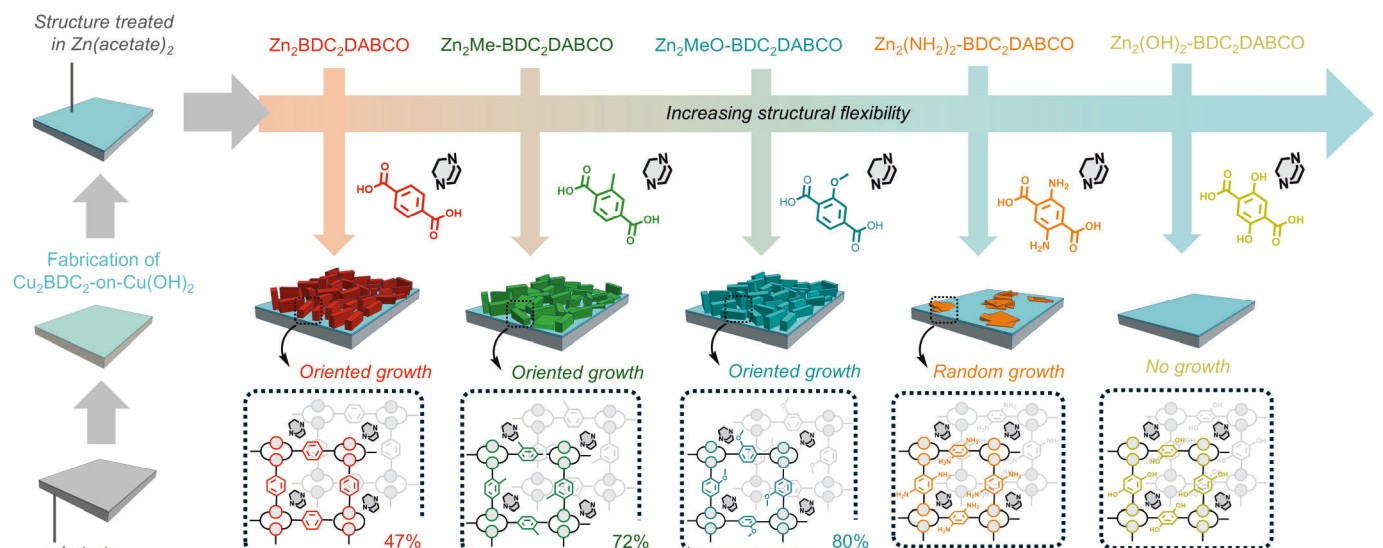

**Fig. 2 | Effect of ligand functionalization on the heteroepitaxially grown zinc-based MOF films.** A stepwise fabrication process is shown on the left, where first the Cu$_2$BDC$_2$-on-Cu(OH)$_2$ structure[35] is grown on substrates and subsequently treated with Zn(acetate)$_2$ to promote the Zn-MOF growth. Various Zn$_2$L$_2$DABCO structures with increasing structural flexibility were produced using DABCO and differently functionalized BDC linkers (L = H$_2$BDC, H$_2$Me-BDC, H$_2$MeO-BDC, H$_2$(NH$_2$)$_2$-BDC and H$_2$(OH)$_2$-BDC), represented from left to right. The resulting structures comprising oriented growth were Zn$_2$BDC$_2$DABCO, Zn$_2$Me-BDC$_2$DABCO and Zn$_2$MeO-BDC$_2$DABCO, whilst Zn$_2$(NH$_2$)$_2$-BDC$_2$DABCO showed random growth, and no growth was obtained for Zn$_2$(OH)$_2$-BDC$_2$DABCO. For the structures displaying oriented growth, the preferred orientation in the out-of-plane direction is given in percent.

these photo-responsive Zn-MOF films when CO$_2$ molecules are present, allowing the reversible uptake and release of CO$_2$ upon irradiation with 365 and 450 nm. Therefore, the experimental approaches delineated herein offer viable methods that can be readily applied to investigate other stimuli-responsive MOF film structures in various gas adsorption configurations.

## Results

### Growth of the MOF film systems

The morphology and growth of Zn$_2$L$_2$DABCO film structures were characterized by SEM and GIWAXS measurements. In Fig. 3a–d, the top-view SEM micrographs are displayed with larger sample areas provided in Figure S2a–c, from which the average crystal sizes were deduced. The lateral size of the Zn$_2$BDC$_2$DABCO crystallites was determined to reach up to ~0.65 μm in length with a width of ~0.25 μm (Fig. 3a). Similarly, the Zn$_2$Me-BDC$_2$DABCO structure displays a comparable morphology where the lateral crystallite size was evaluated to reach ~1.9 μm with a width of ~0.52 μm (Fig. 3b). The Zn$_2$MeO-BDC$_2$DABCO crystallites were found to be significantly different in their size with smaller crystallites reaching laterally ~1.8 μm with a width of ~0.9 μm, whilst others range up to ~4 μm and ~2 μm, respectively (Fig. 3c). The crystallite size could not be determined for the amino functionalized Zn-MOF system, which exhibits a platelet-like morphology as envisioned by the SEM micrograph in Figs. 3d and S2e, f. We ascribe this to the growth of the Zn$_2$(NH$_2$)$_2$-BDC$_2$ structure lacking the pillaring DABCO linker, the latter being crucial for three-dimensional growth of the Zn-MOF (vide infra, Figs. S3–4). Moreover, no conversion to the Zn$_2$(OH)$_2$-BDC$_2$DABCO could be accomplished because of a complete detachment of the substructure from the substrate. This is attributed to the increased redox activity of H$_2$(OH)$_2$-BDC and higher acidity of the methanolic H$_2$(OH)$_2$-BDC/DABCO conversion solution (pH ~ 3–4) compared to the BDC/DABCO solution (pH ~ 5–6, see ESI, chapter 1)[44].

The cuboid-like morphology of the Zn$_2$BDC$_2$DABCO film system was already reported previously along with the crystallographic alignment of its structure[33]. Herein, we confirmed the crystallinity and growth of the Zn-MOF film structures by GIWAXS measurements with the scattering curves shown in Fig. 3e. Following the heteroepitaxial growth concept, the crystal lattice parameters of the upper Zn-MOF structure (bulk Zn$_2$BDC$_2$DABCO, *P4/mmm*, $a = b = 10.95$ Å, $c = 9.61$ Å)[43] are expected to match the lower substructure, Cu$_2$BDC$_2$-on-Cu(OH)$_2$ (*P4*, $a = c = 10.61$ Å, $b = 5.80$ Å)[35]. Moreover, Henke et al. demonstrated[39] that the addition of substituents in position 2 and 5 of the BDC linker results in isostructural growth for Zn-MOFs.

Accordingly, we expected the functionalized films to be hetero-epitaxially grown along the entire film in the in-plane and out-of-plane direction, as already reported previously for the Zn$_2$BDC$_2$DABCO film system[33]. On fact, for Zn$_2$BDC$_2$DABCO, the $a = b$ axis aligns to the $a$-axis of the lower Cu$_2$BDC$_2$-on-Cu(OH)$_2$ substructure (see Fig. S5a)[35], thus ensuring a minimum lattice mismatch (Table S1-2)[45]. The GIWAXS curves for Zn$_2$Me-BDC$_2$DABCO and Zn$_2$MeO-BDC$_2$DABCO were found to closely match the one of Zn$_2$BDC$_2$DABCO (Fig. 3e). Detailed analysis of the in-plane and out-of-plane pattern provided in Fig. S5b, c confirms the isostructural growth principle. For Zn$_2$MeO-BDC$_2$DABCO, the (*hk*0) plane is strongly pronounced in the out-of-plane direction thus being perpendicular to the substrate, whilst the (00*l*) plane orients preferentially in the in-plane direction (Fig. S5b). Azimuthal angle dependence measurements confirmed a crystal alignment in which the $a$-axis for all three Zn$_2$L$_2$DABCO systems matches the $a$-axis of the Cu$_2$BDC$_2$-on-Cu(OH)$_2$ substructure, and orthogonally to that the $c$-axis aligns with the $b$-axis, respectively (Fig. S5d–f). For Zn$_2$MeO-BDC$_2$DABCO and Zn$_2$Me-BDC$_2$DABCO this alignment is supported by a low lattice mismatch in the $a$-axis direction (3.3% and 2.3%, see Table S2)[45]. Considering the lattice parameters evaluated from the GIWAXS pattern with the results provided in Table S1, the lattice mismatch in $c$-axis direction of the upper Zn-MOF structures reaches about 19%. Interestingly, the Zn$_2$BDC$_2$DABCO structure comprises a second alignment by which the $a$- and $c$-axes are rotated in the in-plane direction by 90° (Fig. S5d). As both the methyl and methoxy-functionalized Zn-MOF structure lack this flip in alignment, the absence of this orientation is attributed to the bulkier functional groups and thus larger lattices as evidenced by an increase in the $a$ and $b$-axis parameters (see Table S2), which could be directing the alignment of the top layer Zn$_2$L$_2$DABCO structure. Because of this lattice orientation for both Zn$_2$MeO-BDC$_2$DABCO and Zn$_2$Me-BDC$_2$DABCO with respect to the lower Cu$_2$BDC$_2$ structure, both systems result in

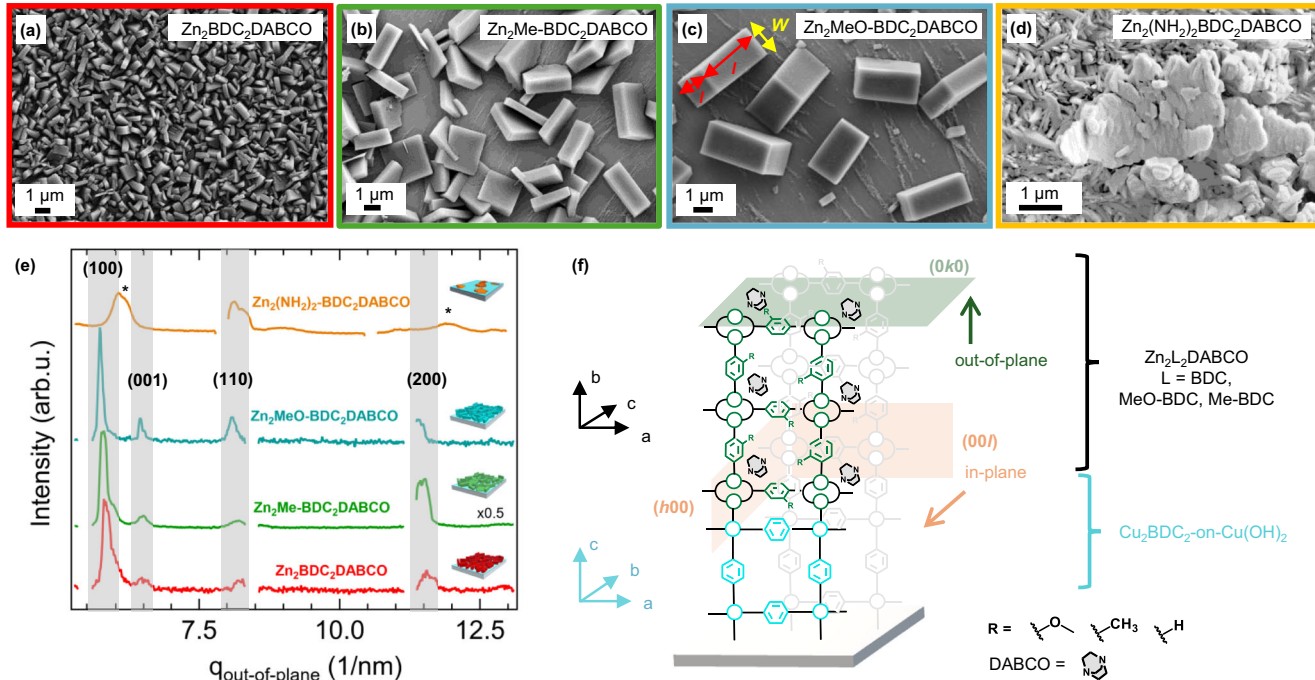

**Fig. 3 | SEM micrographs and GIWAXS pattern of the Zn-MOF films. a** SEM micrographs of $Zn_2BDC_2DABCO$, **b** $Zn_2Me-BDC_2DABCO$, **c** $Zn_2MeO-BDC_2DABCO$, **d** $Zn_2(NH_2)_2-BDC_2DABCO$. The length (l) and width (*w*) of the crystallites are indicated by red and yellow arrows, respectively in (**c**). **e** GIWAXS pattern evaluated for the out-of-plane direction for $Zn_2(NH_2)_2-BDC_2DABCO$, $Zn_2MeO-BDC_2DABCO$, $Zn_2Me-BDC_2DABCO$ and $Zn_2BDC_2DABCO$. The reflections (100), (001), (110) and (200) corresponding to the Zn-MOF lattice are indicated by the areas highlighted in gray. The asterisks denote reflections related to the $Cu_2BDC_2$-on-$Cu(OH)_2$ substructure[35]. **f** Schematics for the heteroepitaxial growth of the $Zn_2L_2DABCO$ structures along the in-plane and the out-of-plane directions (L = BDC, Me-BDC and MeO-BDC, side-views are provided in Fig. S5, g). The (00 *l*) and (*h*00) planes align in-plane (orange highlighted areas), whilst the (0*k*0) plane (green highlighted area) orients in the out-of-plane direction being parallel to the substrate. Source data are provided as a Source Data file.

being more flexible compared to the $Zn_2BDC_2DABCO$ system (see ESI for more details in chapter 6 and Fig. S5g). This result confirms that although the three Zn-MOF structures are grown isostructural, the functionalization of the Zn-MOF structures enforces a flip in epitaxial alignment, whilst an increase of the lattice constants causes a growing lattice mismatch with respect to the smaller substructure.

On the contrary, GIWAXS results revealed that the amino-functionalized structure lacks reflections which are characteristic for the Zn-MOF system (Fig. 3e). A more detailed analysis of the results is provided in Fig. S4, showing no preferential alignment in the in-plane or out-of-plane direction. This finding indicates that this system grows in a non-isostructural manner. Moreover, IR measurements designate the absence of vibrational modes related to the DABCO linker molecule[46] indicating that the $Zn_2(NH_2)_2-BDC_2$ structure lacks this pillaring linker which is crucial for the three-dimensional growth of the structure (Fig. S3). This result is supported by SEM micrographs where instead of a cuboid-geometry, a platelet-like growth occurred (Figs. 3d and S2e, f). We attribute this lack of isostructural and oriented growth to be a result of the amino substituents being in close proximity, which increases the organic-organic intraframework steric repulsion and, therefore, hinders MOF growth[44].

For the isostructural and heteroepitaxial Zn-MOF films, orientation analysis showed that 80% of the $Zn_2MeO-BDC_2DABCO$ crystallites and 72% for $Zn_2Me-BDC_2DABCO$ orient in out-of-plane direction with their long axis being perpendicular to the substrate (see ESI chapter 6−7 and Fig. S6). This is in good agreement with results obtained from SEM micrographs where a preferential out-of-plane orientation of the crystallites was observed (see Figs. 3 and S2a−c). A lower degree of orientation can be explained by the presence of crystallites that align *i.e.*, face-down (see Fig. S2, d), as envisioned exemplarily in the SEM micrographs in Fig. 3b. For comparison, the degree of orientation for $Zn_2BDC_2DABCO$ crystallites was determined with 47%, which is in good

agreement to reported data[33]. Yet, the strong preferential alignment of the crystallites is further corroborated by the presence of the (101) reflection in the in-plane direction (see Fig. S5). Considering these results, the heteroepitaxial Zn-MOF growth with respect to the $Cu_2BDC_2$-on-$Cu(OH)_2$ substructure for the isostructural systems is schematically outlined in Fig. 3f.

## Low-pressure $CO_2$ uptake in heteroepitaxial Zn-MOF films

The free $CO_2$ gas molecule exhibits four fundamental vibrational modes, where only the symmetric stretching ($v_{1-CO2}$) is Raman active, while the doubly degenerated bending ($v_{2-CO2}$) and the antisymmetric stretching vibrations ($v_{3-CO2}$) are both infrared active[47−49]. For free $CO_2$ in gas phase, these modes are at 1388.3 cm$^{-1}$ ($v_{1-CO2}$), 667.3 cm$^{-1}$ ($v_{2-CO2}$) and 2349.3 cm$^{-1}$ ($v_{3-CO2}$)[27]. In MOFs, $CO_2$ molecules are weakly adsorbed, as indicated by vibrational modes around 640 and 650 cm$^{-1}$[27,31,50], which herein are denoted as $v_{CO2-ads}$. Moreover, if $CO_2$ is sufficiently adsorbed by the MOF system, expansion of the pores and eventually of the crystalline lattice will give rise to structural modulations visible in its X-ray pattern.

We investigated the structural changes and arising interactions by means of IR spectromicroscopy, GIWAXS and QCM-D (see Fig. 1), to evaluate the response of the heteroepitaxial Zn-MOF films towards $CO_2$. The combination of these techniques is crucial, as the presence of adsorbed $CO_2$ molecules over free $CO_2$ can be distinguished mainly via IR absorption spectroscopy[27], while GIWAXS measurements complement the information related to structural changes arising from crystal lattice distortions[33] occurring upon $CO_2$ uptake, and the adsorbed amount is quantified by QCM-D measurements. Gravimetric measurements of the respective Zn-MOF film layers are summarized in Table S3.

To this aim, in a first step, IR measurements were conducted on the Zn-MOF films in the absence of $CO_2$ with the spectral features

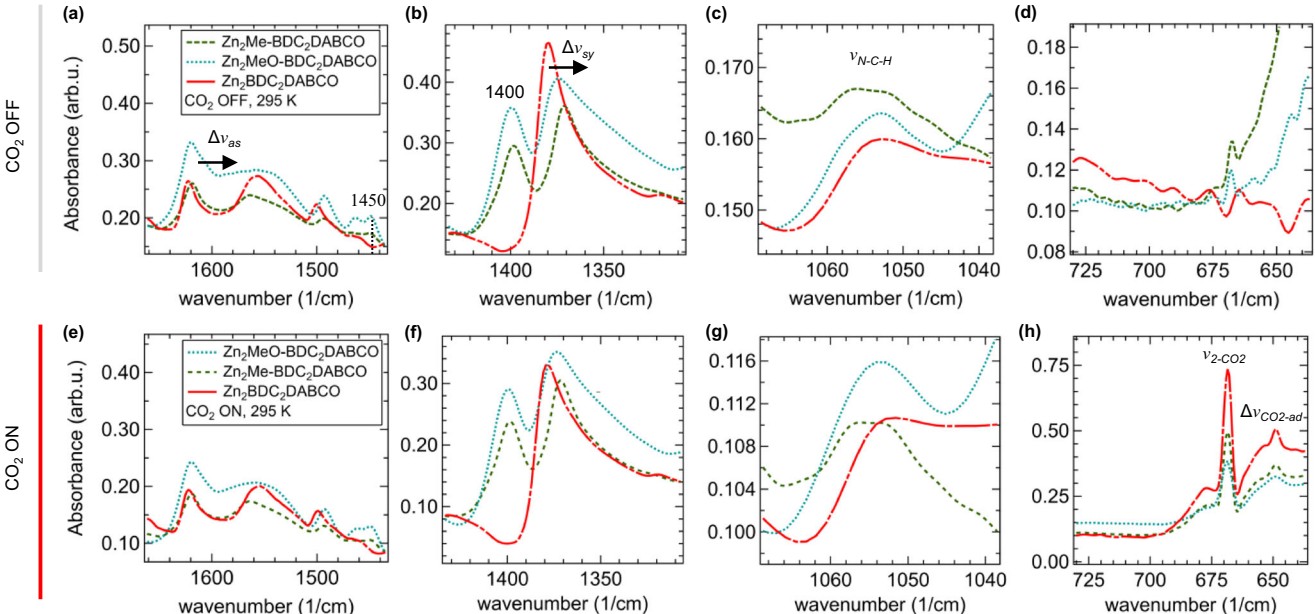

**Fig. 4 | FT-IR spectra of the Zn-MOF films in absence ($CO_2$ OFF) and presence ($CO_2$ ON) of $CO_2$ flow.** $Zn_2MeO$-$BDC_2DABCO$ (blue dotted), $Zn_2Me$-$BDC_2DABCO$ (dark green dashed) in comparison to the $Zn_2BDC_2DABCO$ film system (no BDC-linker functionalization, red dash point). Zoom-in on (**a**) the asymmetric ($v_{as}$) and, (**b**) the symmetric carboxylate vibrations ($v_{sy}$) and (**c**) the vibrational mode attributed to the N-C-H moiety of the DABCO linker ($v_{N\text{-}C\text{-}H}$). At 1400 $cm^{-1}$ and 1450 $cm^{-1}$ vibrational modes attributed to the methyl and methoxy functionality are indicated in (**a**, **b**). **d** Purging by $N_2$ revealed a negligible signal related to non-adsorbed $CO_2$ molecules ($v_{CO2}$). Upon exposure to $CO_2$ the characteristic regions are shown in (**e**–**h**), respectively (see main text for discussion), where $\Delta v_{CO2\text{-}ad}$ relates to adsorbed $CO_2$. The $\Delta v$ symbols refer to shifts between the functionalized Zn-MOF films (see main text) and the arrows denote the direction of the shift. Same color coding for all images. Source data are provided as a Source Data file.

provided in Fig. 4a–d ($CO_2$ OFF). Since both $Zn_2Me$-$BDC_2DABCO$ and $Zn_2MeO$-$BDC_2DABCO$ showed isostructural growth to the non-functionalized $Zn_2BDC_2DABCO$ film system (vide supra), a direct comparison of the observed features is reasonable[33,51]. The regions of interest for the Zn-MOF structures are related to the asymmetric ($v_{as}$, Fig. 4a) and symmetric ($v_{sy}$, Fig. 4b) carboxylate vibrations, as well as modes attributed to the deformation of the N-C-H moiety in the DABCO linker ($v_{N\text{-}C\text{-}H}$, Fig. 4c)[51,52]. These modes are considered in order to unravel structural differences within the Zn-MOF films. Comparing the functionalized Zn-MOF to $Zn_2BDC_2DABCO$, the asymmetric carboxylate vibration located at 1622 $cm^{-1}$ is red-shifted by $\Delta v_{as}$ = -3.8 $cm^{-1}$ for $Zn_2Me$-$BDC_2DABCO$ and by $\Delta v_{as}$ = −2.8 $cm^{-1}$ for $Zn_2MeO$-$BDC_2DABCO$ (Fig. 4a). This trend is also found for the symmetric carboxylate vibration located at 1379 $cm^{-1}$ for $Zn_2BDC_2DABCO$, which for $Zn_2Me$-$BDC_2DABCO$ is shifted by $\Delta v_{sy}$ = −8.4 $cm^{-1}$ and by $\Delta v_{sy}$ = −5.1 $cm^{-1}$ in the case of $Zn_2MeO$-$BDC_2DABCO$ (Fig. 4b), whilst for the N-C-H moiety of DABCO only a modulation of the vibrational band is observed (Fig. 4c). These differences support earlier GIWAXS observations (vide supra, Fig. 3e), where the functional groups were found to modify the Zn-MOF structure to some extent. The presence of the methoxy and methyl group in the BDC linker was further confirmed by the appearance of the -$OCH_3$ and -$CH_3$ bending vibrations at ~1450 $cm^{-1}$ and ~1400 $cm^{-1}$, respectively (Fig. 4a, b)[38,53].

We subsequently exposed the Zn-MOF films to low-pressure $CO_2$ flow, while performing QCM-D measurements to quantify the amount of adsorbed $CO_2$ (Fig. S7). Results showed an immediate response of the Zn-MOF films towards $CO_2$ with strong and rapid gas adsorption, much in contrast to the responses obtained for the bare crystal or $Cu_2BDC_2$ substructure (see ESI chapter 8 and Fig. S7a). Importantly, the methyl and methoxy functionalities lead to a significantly higher $CO_2$ uptake with 0.46 ± 0.03 µg/cm² and 0.54 ± 0.03 µg/cm², respectively, when compared to the non-functionalized $Zn_2BDC_2DABCO$ film system (0.14 ± 0.01 µg/cm²). Based on the mass of the functionalized Zn-MOF films on the QCM-D crystal, the $CO_2$ adsorption capacities are

0.17 $mmol_{CO2}/g_{Zn2Me\text{-}BDC2DABCO}$ and 0.40 $mmol_{CO2}/g_{Zn2MeO\text{-}BDC2DABCO}$ (Table S3). Considering the low-pressure conditions at ambient temperature, these adsorption capacities outperform some MOF-based adsorbents operating at higher $CO_2$ pressure[3]. Moreover, irrespective on the functionality, the amount of adsorbed $CO_2$ was reversibly released by the Zn-MOF structures upon purging with nitrogen ($CO_2$ OFF, Fig. S7).

We further performed infrared spectroscopy and GIWAXS measurements to unravel the chemical and structural response occurring upon $CO_2$ adsorption within the films. The regions of interest in the IR spectra of the functionalized film structures upon exposure to $CO_2$ at 295 K ($CO_2$ ON) are displayed in Fig. 4e–h. The appearance of a vibrational mode at $v_{2\text{-}CO2}$ = 668 $cm^{-1}$ is related to the free $CO_2$ molecule, which matches literature reports[31] and indicates saturation of the sample compartment by $CO_2$. Closer inspection of this region reveals a broad peak located at $v_{CO2\text{-}ad}$ = 648 $cm^{-1}$ (Fig. 4h). Such a mode has been described in the case of ZIF-8[31] and MIL-53(Al)[27] and was attributed to the adsorption of $CO_2$ under high pressures. The presence of this mode coupled with gravimetric results obtained with QCM-D demonstrate that the oriented Zn-MOF films readily adsorb $CO_2$.

To investigate interactions between adsorbed $CO_2$ and the Zn-MOF host structure, we leveraged the flexibility of the frameworks. Upon guest molecule uptake, $Zn_2BDC_2DABCO$ and similar isostructural analogues[39,46,54,55] undergo a structural transformation, owed to their pillared-layered assembly, in which bulk systems reversibly transform between a large-pore (LP) and narrow-pore phase (NP)[41]. This process commences upon gas ad- or desorption and is often termed "breathing" because of the reversible expansion or contraction of the pores[56,57]. This transition alters the proximity of organic linker molecules and thus the chemical environment in the MOF pores, affecting adsorbed molecules and causing shifts of vibrational bands[50,58]. However, the LP-to-NP transition can also be triggered by stimuli such as light (vide infra) or temperature[41]. At low temperatures the NP phase becomes stabilized, whilst an increase of temperatures

**Table 1 | Evaluated spectral features for the $Zn_2L_2DABCO$ films (L = MeO-BDC, Me-BDC, BDC) prior and after $CO_2$ exposure ($CO_2$ OFF/ON) at a temperature difference of $\Delta T = 95$ K**

| | | $\Delta\nu_{as}$ | $\Delta\nu_{sy}$ | $\Delta\nu_{N\text{-}C\text{-}H}$ | $\Delta\nu_{CO2\text{-}ad}$ |
|---|---|---|---|---|---|
| $Zn_2$MeO-BDC$_2$DABCO | $CO_2$ OFF | −3.8 | +1.2 | n.d. | ---- |
| | $CO_2$ ON | −3.8 | 0 | +1.2 | +1.9 |
| $Zn_2$Me-BDC$_2$DABCO | $CO_2$ OFF | −1.2 | 0 | +1.2 | ---- |
| | $CO_2$ ON | −1.2 | 0 | 0 | +1.2 |
| $Zn_2$BDC$_2$DABCO | $CO_2$ OFF | 0 | 0 | n.d. | ---- |
| | $CO_2$ ON | −3.2 | +1.9 | n.d. | 0 |

The (-) sign denotes a red-shift and (+) a blue-shift of the vibrational modes $\Delta\nu_{sy}$ (symmetric carboxylate vibrations), $\Delta\nu_{as}$ (asymmetric carboxylate vibrations), $\Delta\nu_{N\text{-}C\text{-}H}$ (vibrational mode attributed to the N-C-H moiety of the DABCO linker) and $\Delta\nu_{CO2\text{-}ad}$ (adsorbed $CO_2$). (for spectra see Figs. S8–9).
n.d. not determined due to increased background or resolution limit.

promotes the structure to enter its LP phase[59]. It must be noted, that the film structures investigated herein are heteroepitaxially grown and thusly, a lowering in temperature is expected to evoke structural changes only to a limited extent[33].

Following this, we conducted low-temperature studies to determine if changes in the Zn-MOF environment affect adsorbed $CO_2$ molecules. To this aim, it is particularly crucial to differentiate features arising due to temperature effects or the adsorption process itself. Thus, in a first step, we performed IR spectromicroscopy measurements on the film structures in the absence of $CO_2$ at 295 K, 240 K and 200 K ($CO_2$ OFF). The most pronounced vibrational differences were observed when comparing the spectra taken at 295 K and 200 K ($\Delta T = 95$ K, see Figs. S8–9) with the spectral changes summarized in Table 1. Without $CO_2$, the $Zn_2$MeO-BDC$_2$DABCO structure experiences a considerable red-shift with $\Delta\nu_{as} = -3.8$ cm$^{-1}$ and weaker blue-shifts of $\Delta\nu_{sy} = +1.2$ cm$^{-1}$ upon the reduction of temperature by $\Delta T = 95$ K, along with a slight modulation of $\nu_{N\text{-}C\text{-}H}$. $Zn_2$Me-BDC$_2$DABCO showed subtle changes with $\Delta\nu_{as} = -1.2$ cm$^{-1}$ and $\Delta\nu_{N\text{-}C\text{-}H} = +1.2$ cm$^{-1}$ (Fig. S8), while $Zn_2$BDC$_2$DABCO experienced no significant changes (Fig. S9). These differences are attributed to enhanced structural flexibility that is supported with QCM-D results showing a 30% higher $CO_2$ uptake for functionalized Zn-MOF films.

We then measured FT-IR spectra upon $CO_2$ adsorption (see Figs. S8–9). Both $Zn_2$MeO-BDC$_2$DABCO and $Zn_2$Me-BDC$_2$DABCO retained the red-shift of the asymmetric carboxylate mode with $\Delta\nu_{sy} = -3.8$ and $-1.2$, respectively ($CO_2$ ON, Table 1). Only the methoxy-functionalized structure showed a blue-shift for the N-C-H mode of $\Delta\nu_{N\text{-}C\text{-}H} = +1.2$ upon $CO_2$ uptake, which is indicative to the increased structural flexibility (see Fig. S8). While $Zn_2$BDC$_2$DABCO exhibited significant changes for $\Delta\nu_{as}$ and $\Delta\nu_{sy}$ (see Fig. S9), only the functionalized Zn-MOF structures showed a blue-shift of the $\Delta\nu_{CO2\text{-}ad}$ mode with decreasing temperature, which was most pronounced for the methoxy-functionality (Table 1). These findings strongly support the increase in structural flexibility caused by the linker functionalization as well as the presence of adsorbed $CO_2$ molecules within the MOF lattice, which interact with the functional groups of the BDC linkers. Closer inspection of the $CO_2$ OFF/ON infrared spectra at 295 K for $Zn_2$MeO-BDC$_2$DABCO revealed a blue-shift of the vibrational band attributed to the -OCH$_3$ moiety by $\Delta\nu_{\text{-}OCH3} = +1.2$ cm$^{-1}$ (295 K, Fig. S10a). Such a feature could not be disclosed for $Zn_2$Me-BDC$_2$DABCO, which showed only a slight modulation in this region (Fig. S11). Based on these findings, the methoxy functionality promotes the strongest interaction with the $CO_2$ adsorbate[55] that is entirely absent for the non-functionalized $Zn_2$BDC$_2$DABCO film system[44].

We then investigated the structural changes of the Zn-MOF film systems under low-pressure $CO_2$ flow by GIWAXS measurements. To evaluate the extent of induced changes, patterns were taken in the

beginning ($CO_2$ start) and after $CO_2$ saturation ($CO_2$ end) with results shown in Fig. 5a–c and Fig. S12. The most pronounced structural changes were observed for the (100) out-of-plane (OP) and (001) in-plane (IP) direction. The $CO_2$ OFF/ON experiments were performed at room temperature, because of our focus set on $CO_2$ uptake at near-ambient conditions. The $Zn_2$MeO-BDC$_2$DABCO again represented the most flexible structure, as results indicated from earlier QCM-D and IR measurements. GIWAXS results show that the system expands upon $CO_2$ uptake in the (100)$_{OP}$ direction by $\Delta q_{\text{out-of-plane}}(100) = 0.01$ nm$^{-1}$ that orients along the directions of the OCH$_3$-BDC linkers (Fig. 5a, *left*), while contracting in the (001)$_{IP}$ direction by $\Delta q_{\text{in-plane}}(001) = 0.014$ nm$^{-1}$, representing the direction along the DABCO linker (Fig. 5a, right). These changes were found fully reversible upon the desorption of $CO_2$. Similarly, $Zn_2$Me-BDC$_2$DABCO showed a reversible structural change, with a slightly stronger expansion in the (100)$_{OP}$ direction by $\Delta q_{\text{out-of-plane}}(100) = 0.015$ nm$^{-1}$ (Fig. 5b). In contrast, $Zn_2$BDC$_2$DABCO exhibited irreversible changes upon $CO_2$ exposure (Fig. 5c). The observed lattice distortions for the Zn-MOF films during $CO_2$ uptake are schematically depicted in Fig. 5d–f. We conclude that the heteroepitaxial film structures expand along the out-of-plane direction upon $CO_2$ adsorption, whereas only $Zn_2$MeO-BDC$_2$DABCO experiences a shrinkage in the in-plane direction. This difference in behavior is attributed to an increase in structural flexibility owed to the crystal lattice alignment that is also related to the linker functionalities (vide supra, Fig. 3e)[44,60], which further increase the amount of adsorbed $CO_2$ stems due to interactions arising between the adsorbate and the functionalized Zn-MOF structure.

## Structural photo-responsivity under $CO_2$ load

We further explored the incorporation of photo-active azobenzene molecules within the flexible Zn-MOF film pores, which may provide a promising pathway to remotely trigger the $CO_2$ uptake and release using LED light as an energetically efficient stimulus[42,43]. To investigate whether the heteroepitaxial Zn-MOF film structures respond to light under low-pressure $CO_2$ conditions, we first introduced a photo-active molecule[33] using the functionalized $Zn_2L_2DABCO$ film systems (L = Me-BDC and MeO-BDC). Successful azobenzene uptake was confirmed by infrared and UV-Vis spectroscopic measurements (Fig. 6a, b), while QCM-D measurements revealed an increase in the total film mass related to the azobenzene uptake (Table S3).

In $Zn_2$Me-BDC$_2$DABCO, both *trans*- and *cis*-azobenzene conformers are present as evidenced by the vibrational bands located at $\nu_{t\text{-}AB} = 686$ cm$^{-1}$ and $\nu_{c\text{-}AB} = 698$ cm$^{-1}$, respectively (Fig. S13a)[43,61]. Conversely, $Zn_2$MeO-BDC$_2$DABCO exhibits predominantly the *trans*-azobenzene conformer indicated by $\nu_{t\text{-}AB} = 688$ cm$^{-1}$ (Fig. S13b). The shift of $\Delta\nu_{t\text{-}AB} = 1.9$ cm$^{-1}$ between the two Zn-MOF structures is attributed to both, to the different chemical environments due to the methyl and methoxy functionalities and the heteroepitaxial growth of the film systems. The latter can induce strained MOF pores due to lattice mismatch, making them rigid and non-responsive to the azobenzene conformer. This behavior has been reported previously for hetero-epitaxial $Zn_2$BDC$_2$DABCO films, where rigid MOF pores retain a portion of the azobenzene molecules in its *cis*-conformer, whilst only a certain percentage relaxes to its *trans*-isomer[33]. The arousal of strained pores in the $Zn_2$Me-BDC$_2$DABCO film structure is supported by the strong $\nu_{c\text{-}AB}$ mode and only a modest isomerizing fraction (13%, vide infra). The azobenzene loading level was determined from UV-Vis measurements with ~0.7 azobenzene molecules per MOF pore (see ESI for calculation, Table S4). For the $Zn_2$MeO-BDC$_2$DABCO film, the level of azobenzene loading accounts solely for ~0.2 that is considerably less compared to the methyl-functionalized structure (Table S4). Here, mainly the $\nu_{t\text{-}AB}$ mode is present being broad in its peak width, which, considering the low azobenzene loading, hampers a clear statement on the presence of strained pores (Fig. S13). GIWAXS measurements confirmed the persistence of crystallinity and heteroepitaxy upon

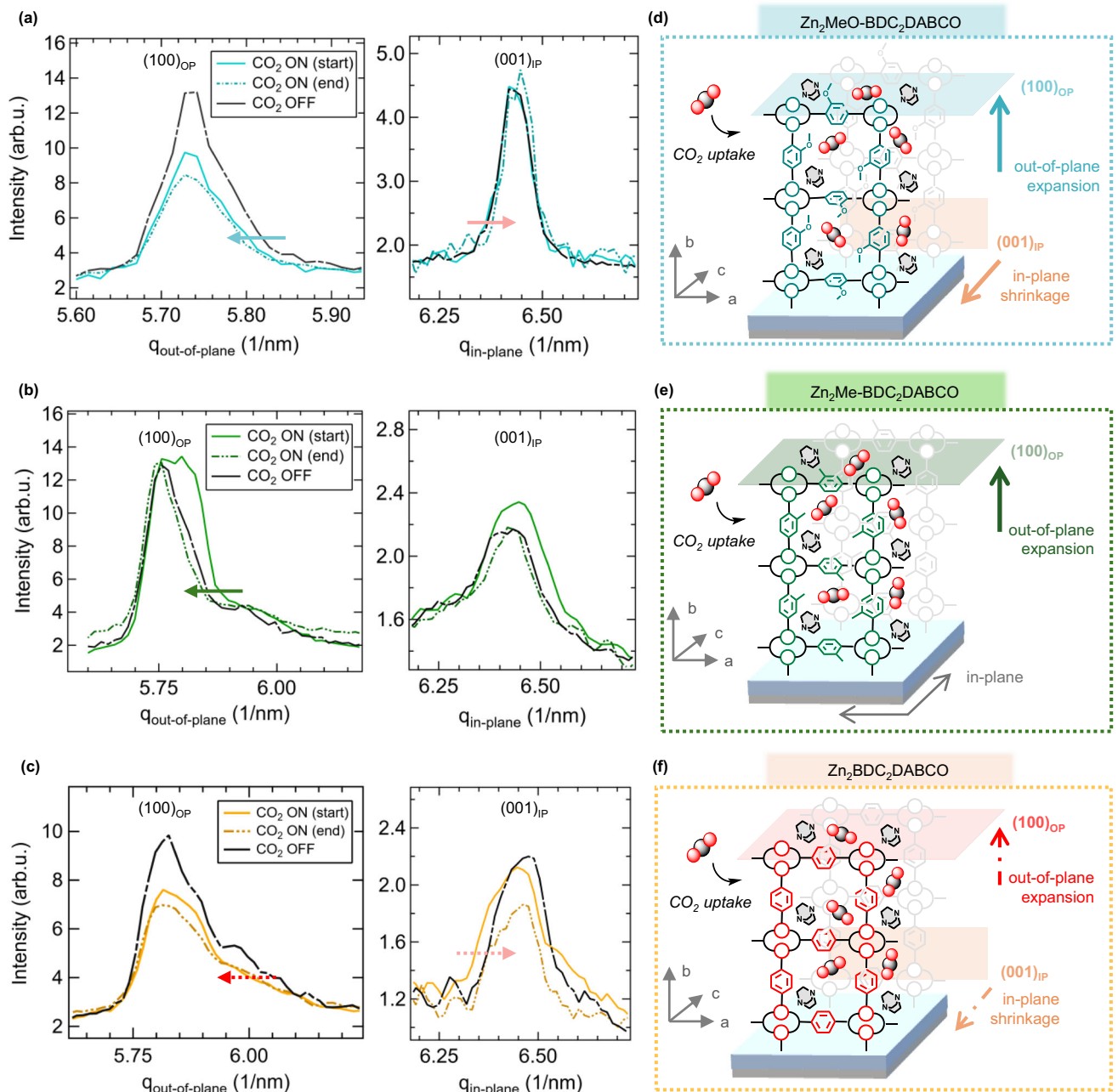

**Fig. 5 | GIWAXS results of the heteroepitaxial Zn₂L₂DABCO film structures, with L = BDC, MeO-BDC and Me-BDC upon CO₂ uptake and release (295 K).** Measurements were taken in the beginning (CO₂ start), after sufficient CO₂ load (CO₂ end) and after subsequent purging with N₂ (CO₂ OFF) considering the out-of-plane (OP) and the in-plane (IP) direction. **a** Zn₂MeO-BDC₂DABCO shows an expansion of the (100)$_{OP}$ and shrinkage of the (001)$_{IP}$ reflection as indicated by arrows. **b** Similarly, Zn₂Me-BDC₂DABCO experiences an expansion along the (100)$_{OP}$

reflection, whilst (001)$_{IP}$ remains silent. **c** Zn₂BDC₂DABCO shows only a subtle expansion along (100)$_{OP}$ and contraction for the (001)$_{IP}$ reflection as indicated by arrows. Schematic description of the observed features upon CO₂ uptake are depicted in **d** for Zn₂MeO-BDC₂DABCO, **e** Zn₂Me-BDC₂DABCO and **f**) for Zn₂BDC₂DABCO (solid arrow = reversible, gray arrow = no changes, punctuated arrow = subtle, irreversible change). Source data are provided as a Source Data file.

azobenzene infiltration (Fig. S14a–c). Here, Zn₂MeO-BDC₂DABCO/AB experiences a pronounced structural shrinkage, with $\Delta d = 0.22$ Å for the (100) reflection (Fig. S14b), compared to $\Delta d = 0.08$ Å for Zn₂Me-BDC₂DABCO/AB (Fig. S14b). This finding supports the increased structural flexibility due to the methoxy functionality which is in line to the findings discussed earlier. Moreover, the contraction of the heteroepitaxial structures indicates the arousal of strong interactions between the Zn-MOF and the azobenzene molecule upon its uptake within the MOF pores[33,43]. QCM-D results show a significant decrease in CO₂ uptake (Fig. S15a, b) with the adsorbed amount in the functionalized Zn-MOF/AB accounting for $0.04 \pm 0.01 \, \mu g/cm^2$ (Fig. 6e). These

data show that the azobenzene molecules still leave sufficient space inside the MOF for CO₂ uptake[29] considering the respective azobenzene loading levels (see ESI, chapter 11).

In the next step, the photo-response of the infiltrated Zn-MOF films in the presence of CO₂ was investigated by IR spectromicroscopy and QCM-D measurements (Fig. 1b, c). To this aim, a wavelength of 365 nm isomerizes azobenzene to its *cis*-conformer, whilst 450 nm relaxes the molecule to the *trans*-isomer as schematically outlined in Fig. 6a[42,43]. The reversible azobenzene photo-isomerization commenced within the Zn-MOF film pores was confirmed by UV-Vis measurements (Fig. 6a, b), and FT-IR measurements (Fig. 6c, d). Notably,

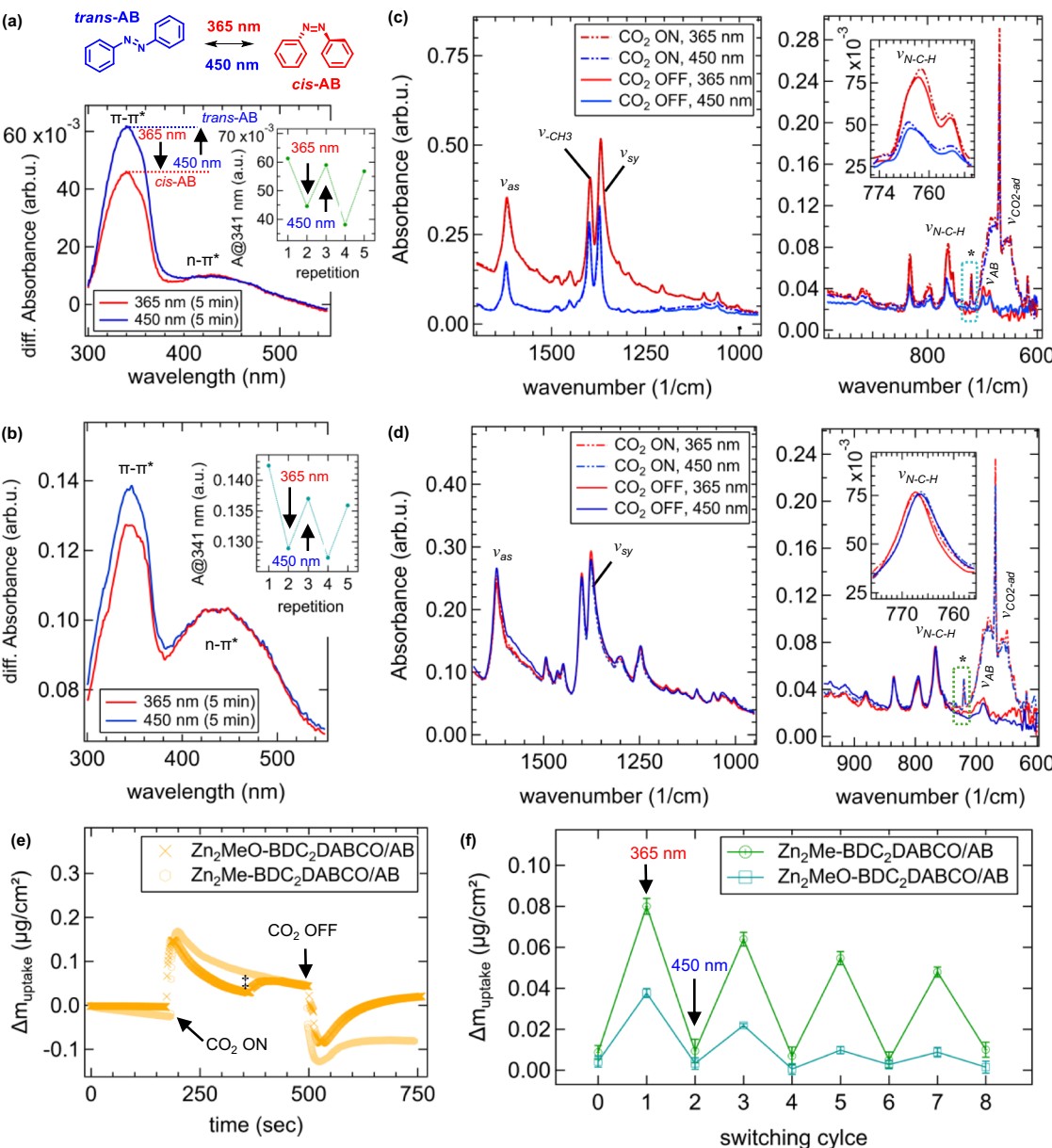

**Fig. 6 | Azobenzene infiltrated Zn-MOF films prior and upon low-pressure CO₂ load. a** UV-Vis spectra show the reversibly induced *trans*-to-*cis* isomerization of azobenzene (see schematic) for Zn₂Me-BDC₂DABCO/AB and **b** Zn₂MeO-BDC₂DABCO/AB using light of 365 nm and 450 nm wavelength. The inset graphs show the change in absorbance at 341 nm after five photo-cycling repetitions. FT-IR spectra show the structural response of **c** Zn₂Me-BDC₂DABCO/AB and, **d** Zn₂MeO-BDC₂DABCO/AB upon the photo-induced azobenzene isomerization prior (solid line spectra) and upon CO₂ low-pressure load (dotted line spectra). The asymmetric ($v_{as}$) and symmetric ($v_{sy}$) carboxylate vibrations, the methyl group vibration ($v_{-CH3}$) and the vibrational mode attributed to the N-C-H moiety of the DABCO linker ($v_{N-C-H}$) are indicated. Asterisks at 720 cm⁻¹ denote the mode ascribed to the

interaction between azobenzene and CO₂. The vibrational modes corresponding to azobenzene ($v_{AB}$) and adsorbed CO₂ ($v_{CO2-ad}$) are also indicated. **e** AB infiltration into the Zn-MOF structures leads to a reduced CO₂ adsorption accounting up to 0.04 ± 0.01 μg/cm². Notably, the methoxy functionality results in a two-step CO₂ uptake process with the second step indicated by a double dagger. Arrows indicate the moment, when the CO₂ pressure load was applied (CO₂ ON) and removed (CO₂ OFF). **f** The photo-triggered CO₂ uptake and release cycles were monitored with the relative CO₂ mass change shown when the Zn-MOFs were irradiated at 365 nm and 450 nm (indicated by arrows). Error bars represent the standard deviation of three independent measurements. Source data are provided as a Source Data file.

the mode related to the C-H bending in azobenzene ($v_{AB}$)[43,61] is superposed both, to the doubly degenerated bending vibrations of CO₂ ($v_{2\text{-}CO2}$) and the mode related to the adsorbed CO₂ portion (vide supra, $v_{CO2-ad}$). Photo-isomerization triggers the LP-to-NP transition in Zn-MOF films (vide supra), which results in a structural response traceable by IR spectromicroscopy[50,58]. This transition alters the accessible pore volume for gases and thus the gravimetric changes related to adsorbed CO₂ were monitored by QCM-D measurements. Results confirmed a successful CO₂ uptake and release triggered upon irradiation of the Zn-MOF films with a significantly stronger response

obtained for the methyl-functionalized system (Fig. 6f). The decrease in sorption capacity after the first photo-switch is related to the lowering of the *trans*-to-*cis* isomerization fraction (Fig. S13c, d).

To unravel the influence of the azobenzene isomerization on the Zn-MOF structure prior and after exposure to CO₂, we initially performed photo-triggered experiments in the absence of CO₂ and repeated the measurement upon CO₂ load. The resulting FT-IR spectra with and without CO₂ are shown in Fig. 6c, d. For Zn₂Me-BDC₂DABCO/AB, about 13% of the azobenzene molecules isomerize, which was determined considering the $v_{AB}$ peak area (Fig. S13a, b). Commencing

the *trans*-to-*cis* transition within the Zn-MOF pores in the absence of $CO_2$ leads to a strong red-shift of the vibrational modes related to the symmetric stretching of the carboxylic group in Me-BDC ($\Delta v_{sy}$ (*trans*-to-*cis*) = −3.2 cm$^{-1}$) and the methyl group ($\Delta v_{-CH3}$ (*trans*-to-*cis*) = −2.6 cm$^{-1}$), which equals the behavior in the presence of $CO_2$ (Fig. 6a). Interestingly, the mode related to the DABCO molecule shifts considerably stronger in the presence of $CO_2$ with $\Delta v_{N-C-H}$ (*trans*-to-*cis*) = +3.2 cm$^{-1}$ ($CO_2$ ON), as it is the case when $CO_2$ is absent ($\Delta v_{N-C-H}$ (*trans*-to-*cis*) = +1.9 cm$^{-1}$, see inset in Fig. 6c). Owed to the strong red-shift, this property indicates that the presence of $CO_2$ supports the $Zn_2Me$-$BDC_2DABCO$/AB structure to enter a slightly more relaxed state in the presence of *cis*-azobenzene[43], which is supported by QCM-D results showing a pronounced photo-triggered $CO_2$ uptake and release. This behavior can be explained considering that upon $CO_2$ exposure, a mode located at 720 cm$^{-1}$ appears for both, $Zn_2Me$-$BDC_2DABCO$/AB and $Zn_2MeO$-$BDC_2DABCO$/AB, which was neither observed in the non-infiltrated systems exposed to $CO_2$ (see Fig. S8d), nor for the azobenzene infiltrated films (Fig. 6c, d). For free azobenzene, strong IR active modes located in the range of 780 and 680 cm$^{-1}$ are attributed to out-of-plane C-H and ring torsion vibrations of *trans*-azobenzene[62]. Moreover, in the case of strong adsorption between azobenzene and other molecules, the sudden appearance of modes designates a change in their dynamic dipole[63] which, in the present case, is related to the interaction between azobenzene and $CO_2$. Based on theoretical calculations for azo-functionalized Zn-MOF structures, such a behavior can be expected since adsorbed $CO_2$ molecules can interact strongly with the nitrogen atoms of *trans*-azobenzene[64]. Notably, $CO_2$ uptake measurements performed by QCM-D revealed a two-step adsorption process in the case of the methoxy-functionality (Fig. 6e), which can be attributed to interactions arising between the *trans*-azobenzene molecule and $CO_2$ (Fig. 6f).

For the $Zn_2MeO$-$BDC_2DABCO$/AB film, only a modulation of the -N-C-H mode upon photo-excitation with $\Delta v_{N-C-H}$ (*trans*-to-*cis*) = −1.3 cm$^{-1}$ was found. This behavior is somehow unexpected since this system was found to be the most flexible based on results discussed earlier. We attribute this finding to be related mainly to the significantly lower azobenzene uptake that is reciprocated by the low $CO_2$ photo-uptake obtained from QCM-D measurements (Fig. 6f), but, on fact, this behavior can also arise because of the slightly larger MOF pores in the case of MeO-BDC (see Table S1). In this case, the azobenzene molecule can freely isomerize without requiring the MOF pore to adapt[65]. For $Zn_2MeO$-$BDC_2DABCO$, IR results denote a subtle structural adaptation during the *trans*-to-*cis* isomerization (Fig. 6d), indicating that the framework responds towards the azobenzene conformer. Closer inspection of the $v_{AB}$ mode upon photo-excitation revealed a significant broadening of the vibrational band implying that the disorder within the structure increases, with 22% of azobenzene molecules isomerizing to its *cis*-conformer (Fig. S13). In contrast, the $Zn_2BDC_2DABCO$/AB film shows no structural changes upon *trans*-to-*cis* azobenzene isomerization (46% of AB molecules)[33] under $CO_2$ load, which supports earlier findings that this system is less flexible (Fig. S16). Considering these findings, the azobenzene infiltrated methyl and methoxy-Zn-MOF films allow to successfully control the $CO_2$ uptake and release simply by photo-triggering the *trans*-to-*cis* azobenzene isomerization (Fig. 6c, d).

## Discussion

Herein, on the example of heteroepitaxially grown $Zn_2L_2DABCO$ films employing a small library of functionalized BDC-linker molecules, we demonstrate that the Zn-MOF films readily adsorb $CO_2$ under low gas pressures. Our findings showcase that despite epitaxial constraints, linker functionalization increases the framework flexibility thus improving the responsive behavior of the $Zn_2L_2DABCO$ films towards guest molecules. The results are based on a combinatory study using GIWAXS, IR spectromicroscopy and QCM-D, which highlights their

beneficence to deduce intermolecular and structural relationships in film systems also when using external stimuli, namely temperature and light. In the first part of our study, we demonstrated reversible $CO_2$ uptake at near-ambient conditions leading to structural distortions of the heteroepitaxial Zn-MOF films with adsorption capacities of 0.17–0.40 mmol $CO_2$/g Zn-MOF. Moreover, when lowering the temperature, the flexibility of the $Zn_2L_2DABCO$ structure was found particularly intriguing as the quantity of adsorbed $CO_2$ is influenced by changes of the structural and chemical environment within the flexible Zn-MOF films. Such a finding is intriguing since the flexibility in heteroepitaxial MOF structures can be severely compromised owed to the need of matching the crystalline lattices for successful film growth. We further examined the responsiveness of the flexible Zn-MOF films under $CO_2$ load in the presence of the photo-active azobenzene molecules within the pores. Results showed that because of the adsorbed azobenzene and $CO_2$ molecules, interactions arise which are accredited to aid the system in adapting to the photo-response. Overall, these findings highlight that the stimuli-responsive behavior of MOF films under $CO_2$ load can be readily explored by the experimental techniques proposed herein. Challenges remain especially when it comes to directly map the spatial distribution of $CO_2$ molecules within the films at the nanoscale. Addressing this limitation could stimulate the development of characterization techniques that are essential for advancing our understanding of transport and storage mechanisms in MOF films. Finally, the methodologies used in this study offer a general conceptual advance for the investigation of structural and molecular changes under *operando* conditions in stimuli-responsive films, which are applicable also to more complex systems such as mixed matrix membranes[9] or assemblies in device configuration[4].

## Methods

### Materials

All chemicals and solvents are available commercially and were used as received without any further purification. The linkers, 2-methoxy-1,4-benzenedicarboxylic acid (H$_2$(MeO-BDC)), 2-methyl-1,4-benzenedicarboxylic acid (H$_2$(Me-BDC)), 2,5-diamino-1,4-benzenedicarboxylic acid (H$_2$((NH$_2$)$_2$-BDC)) and 2,5-di-hydroxyl-1,4-benzene-dicarboxylic acid, (H$_2$((OH)$_2$-BDC)) were purchased from abcr GmbH, while 1,4-benzenedicarboxylic acid (H$_2$BDC), 1,4-diazabicyclo[2.2.2]octan (DABCO) and azobenzene (AB) were obtained from TCI Chemicals. Methanol (MeOH, 99.8%) and absolute ethanol (EtOH, 99.8%) were bought from VWR Chemicals, while acetone was purchased from Avantor™ (99.8%). Preparation of Cu(OH)$_2$ nanobelt films on substrates (1.0 × 1.0 cm$^2$) via a semi-automatic deposition method and their subsequent conversion to Cu$_2$BDC$_2$ was conducted according to the substrate-seeded approach[35]. For IR characterization, Si-wafer substrates were treated with oxygen plasma, while ITO-coated MirrIR glass slides purchased from Kevley Technologies® were employed for the photo-switching experiments. $CO_2$ was bought from Messner in >99.995% purity.

### Pristine Zn-MOF film fabrication ($Zn_2L_2DABCO$)

The Zn-MOF films were grown using the epitaxial Cu$_2$BDC$_2$-on-Cu(OH)$_2$ film system, which was prepared from aligned Cu(OH)$_2$ nanobelts deposited on glass or Si substrates[35]. In the first step, the conversion of Cu$_2$BDC$_2$-on-Cu(OH)$_2$ in a methanolic zinc acetate solution (2.2 mg, 10 mL of MeOH) was done to covalently bind Zn$^{2+}$ that acts as the metal node for the Zn-MOF growth[33]. Subsequently, the converted films were immersed in the methanolic linker solutions containing DABCO and the corresponding organic linker L = 1,4-benzene-dicarboxylate, BDC, 2-methyl-1,4-benzenedicarboxylate, Me-BDC, 2-methoxy-1,4-benzene-dicarboxylate, MeO-BDC, 2,5-diamino-1,4-benzenedicarboxylate, (NH$_2$)$_2$-BDC, 2,5-dihydroxyl-1,4-benzene-dicarboxylate, (OH)$_2$-BDC. (5.0 mg DABCO, 7.0 mg of L), where the

structures were grown for 180 min at 60 °C. Subsequently, the films were rinsed gently with EtOH and dried by a moderate stream of nitrogen.

## Azobenzene infiltration in Zn$_2$L$_2$DABCO

The azobenzene incorporation was conducted according to a vapor assisted procedure[33]. The pristine Zn$_2$L$_2$DABCO film structures (L = BDC, Me-BDC, MeO-BDC) were activated for 15 min at 60 °C and were placed for 60 min in a closed container with droplets of an acetonic azobenzene solution (120 μL, 10 mg/mL).

## GIWAXS/CO$_2$ sorption measurements

GIWAXS measurements of the pristine Zn$_2$L$_2$DABCO structures prior and upon CO$_2$ uptake were conducted at the Austrian SAXS beamline at ELETTRA, Trieste, Italy[66]. Measurements were performed at a sample to detector distance of 480 mm providing a $q$-range from $0.08 < q < 14$ nm$^{-1}$, where $q$ denotes the length of the scattering vector $\left(q = \frac{4\pi}{\lambda}\sin\left(\frac{2\theta}{2}\right)\right)$, $\lambda$ being the wavelength (0.154 nm, 8 keV) and $\theta$ the scattering angle. The beam size was $0.4 \times 0.15$ mm$^2$ ($h$ x $v$). Calibration of the angular scale for the detector was accomplished using silver behenate. The 2D GIWAXS patterns were recorded by a Pilatus3 1 M detector (Dectris Ltd, Baden Switzerland with active area $169 \times 179$ mm$^2$ and a pixel size of 172 μm), where the images were processed by SAXSDOG[67]. Vertical cuts in the out-of-plane direction as well as the horizontal cut in the in-plane direction and radial integration of the GIWAXS pattern were considered for data analysis (Fig. S1). Integrated GIWAXS scattering patterns were processed using IGOR pro (Wavemetrics, Inc., Lake Oswego, OR). The CO$_2$ uptake and release measurements were accomplished using a cylindrical metal chamber[68], where the inlet was connected by a tubing to the CO$_2$ bottle equipped with a needle valve, and the exit to a bubble chamber filled with mineral oil for controlling the flow speed. Kapton windows ensured the passage of the incident and scattered X-ray beam. An outline of the set-up is shown in Fig. 1a. For every film, duplicate measurements were performed at an acquisition time of 60 s and an incident grazing angle of $\alpha_i = 0.2°$.

## IR/CO$_2$ Sorption measurements

Infrared measurements of the Zn$_2$L$_2$DABCO systems were acquired at the Chemical and Life Science branch of the infrared beamline, SISSI-Bio[69] at Elettra Synchrotron in Trieste[70]. Preconditioning of the films was accomplished by purging the systems overnight under continuous nitrogen flow to remove ambient CO$_2$ or water from the structures. To track the dynamic changes of the structures under continuous CO$_2$ flow (0.1 L/min, pressure of 0.1 bar), repeated measurements were collected in transmission mode placing the Zn$_2$L$_2$DABCO films inside the temperature-controlled Linkam FTIR600 stage (Linkam Scientific Instruments, Salfords, UK). The system can work in a flow of gas, and in a temperature range from < −195 °C to 600 °C with a stability of <0.01 °C. It is equipped with two 0.5 mm thick BaF$_2$ windows transparent from UV to 600 nm, ideal for the photo-switching experiments. The experimental set-up is outlined in Fig. 1b, which was purged with nitrogen prior to the measurements to remove moisture and ambient CO$_2$. A reference was collected on each sample before every step of the experiments considering a position on the bare substrate. To track the CO$_2$ uptake and release by the Zn$_2$L$_2$DABCO films, for each condition, five spectra were recorded acquiring 512 scans at 120 kHz scanner speed with a mid-band MCT detector (Infrared Associates, Inc. Stuart, FL, US) using IR synchrotron radiation. CO$_2$ uptake and release experiments were monitored at 295 K, 240 K and 200 K with a spectral resolution of 2 cm$^{-1}$ with a zero-filling factor of 2 before the FFT in the range from 4000 cm$^{-1}$ to 600 cm$^{-1}$. This was accomplished by acquiring spectra of the N$_2$ purged Zn$_2$L$_2$DABCO films (CO$_2$ OFF) at the respective temperatures. Subsequently, the chamber was purged with CO$_2$ until saturation of the characteristic vibrational band at

2349.3 cm$^{-1}$ ($v_{3\text{-}CO2}$) was reached visible in the background spectrum prior acquiring spectra of the CO$_2$ purged Zn$_2$L$_2$DABCO films (CO$_2$ ON). This procedure was repeated at 295 K as triplicates, at 240 K and 200 K for time reasons as duplicates.

## Photo-triggered CO$_2$ uptake

The photo-triggered CO$_2$ uptake experiments were monitored using the infrared spectromicroscopy experimental set-up depicted in Fig. 1b. The study was conducted on the heteroepitaxially grown Zn$_2$L$_2$DABCO film structures (L = BDC, Me-BDC, MeO-BDC) infiltrated with azobenzene (AB). The isomerization of AB was accomplished using portable 365 nm (6 mW) and 450 nm (80 mW) LED diodes. The light sources were fitted to illuminate the sample through the BaF$_2$ window during CO$_2$ uptake measurements (see Fig. 1b, inset).

## Quartz crystal microbalance flow-cell

A proprietary gas flow cell, not available commercially, was designed using FreeCad 0.19 and 3D printed with a Mono 4 K LCD Printer (Anycubic, China) and a white ABS-Like resin (Anycubic, China). On top of the cell a LED setup was glued tightly, equipped with a blue LED (450 nm, Osram, Germany) and a UV LED (365 nm, Würth, Germany). The LEDs were connected to copper foil with conductive paint and glued to a heat transfer tape, which was attached to an aluminium heat sink. Additionally, both LEDs were secured with a silicon glue on the heat transfer tape (Henkel, Germany). Quartz crystal microbalance with dissipation (QCM-D) measurements were realized using the Q-Sense Analyzer (Biolin Scientific), which operates at a resonance frequency of approximately 5 MHz. The experiments were conducted in an open module (QOM 401; sensor QSX 301 gold) at 23 °C covered with the described gas flow cell. The CO$_2$ uptake and release measurements were conducted under continuous CO$_2$ flow (0.1 L/min), after purging of the sample compartment with nitrogen (0.1 L/min) for 2 h containing the Zn$_2$L$_2$DABCO film structure (L = BDC, Me-BDC, MeO-BDC), as well as upon AB infiltration (see ESI, chapter 8). The flow rate of CO$_2$ and nitrogen gas was controlled with calibrated flow controllers (Sierra, Smart Track 100). Triplicate measurements were conducted by recording the CO$_2$ uptake of every MOF layer. Reference measurements were made with uncoated sensors, and the mass of the MOF layers was determined by comparing the resonance frequencies of uncoated and coated crystals in CO$_2$ or nitrogen. For the photo-triggered CO$_2$ uptake, the illumination protocol is given in the ESI (chapter 3).

## SEM Sample preparation and imaging

Morphologies of samples were observed by a scanning electron microscope (Field Emission Scanning Electron Microscope Gemini Column (FEG) ZEISS SIGMA 300; WD = 3.8. 7.5, 7.6 mm; acceleration voltage between 5.00 and 2.00 kV). All samples were coated with Au/Pd prior imaging.

## UV-Vis spectroscopy

Absorbance measurements for the Zn$_2$L$_2$DABCO film structures (L = BDC, Me-BDC, MeO-BDC) prior and after azobenzene infiltration were evaluated using a UV-Vis spectrophotometer (Cary 60, Agilent Technologies). Photo-excitation of the infiltrated azobenzene molecule was accomplished using portable 365 nm (6 mW) and 450 nm (80 mW) LED diodes.

## Data availability

All data needed to evaluate the conclusions in the paper are available within this article. Source data are provided with this paper.

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

## Acknowledgements

S.K. would like to acknowledge CERIC-ERIC for the postdoctoral position granted through its internal project 'INCITE'. The authors acknowledge the CERIC-ERIC consortium for the access to experimental facilities through proposals 20232161 and 20222149, as well as the SISSI-Bio@Elettra for sharing their in-house beamtime. Ing. Andrea Radeticchio is thanked for technical assistance. Supported by TU Graz Open Access Publishing Fund.

## Author contributions

S.K. administered the project, fabricated and characterized the film systems, performed GIWAXS and UV-Vis measurements, created illustrative material, wrote the original draft and conducted formal analysis of the data. B. M., G.B., P.H., C.S., and L.V. substantially assisted in the methodology development, completed all IR measurements. B.S. assisted in the development of the experimental setups for the $CO_2$ uptake/release studies and provided support throughout the measurements. S.D-Z. supplied some starting materials and performed with B.A.-NA SEM imaging. F.L., R.K., and K.S.K invented, designed and produced the QCM-D gas flow cell. H.A. conceptualized the studies, acquired funding and contributed to the scientific discussions to write the original draft.

## Competing interests

The authors declare no competing interests.
