## [Transparent Peer Review file · Nature Communications]

Flexible metal-organic framework films for reversible low-pressure carbon capture and release

Corresponding Author: Professor Heinz Amenitsch

Version 0:

Reviewer comments:

Reviewer #1

(Remarks to the Author)

The current work presents the synthesis of flexible MOF films for low pressure CO₂ adsorption, utilizing three operando methods to study the structural changes of MOFs and their CO₂ adsorption/desorption capacity. The structural changes are induced by CO₂ adsorption or stimuli such as temperature and light. Due to the lack of novelty in the synthesis and characterization, and the insufficient explanation of the CO₂ adsorption/desorption process and structural change mechanism, I do not recommend this paper for publication in Nature Communications. Below are some questions and suggestions for the authors to address.

1. The role of DABCO is unclear. Based on Figure 2 and 3, DABCO acts as free ligands and is not integrated into MOF framework. The role of DABCO needs to be discussed.
2. There is insufficient information on how structural changes occur. Detailed descriptions of the accurate structures before and after the structural changes (e.g. degree of ligand rotation) are necessary.
3. The authors did not explain why reversible structural changes occur in some MOFs but not in others following CO₂ capture and release.
4. Cyclability tests for CO₂ adsorption/desorption are needed.
5. The authors attributed the low CO₂ adsorption capacity in Zn-MOF/AB to structural contraction. However pore blockage by AB can significantly effect CO₂ capture. This part should be discussed in detail.
6. The CO₂ sorption capacity of AB functionalized MOF significantly decreased after 4 cycles, particularly for Zn₂MeO. The authors should provide explanations.
7. The adsorption capacity (mmol/g) of AB functionalized MOFs should be provided for comparison.
8. In figure 6e and S14, the uptake difference from QCM-D is below 0, with some values remaining negative even after reaching the plateau. Since this is closely related to equipment accuracy, the authors should explain.

Reviewer #2

(Remarks to the Author)

Reviewer #3

(Remarks to the Author)

In their manuscript, Klokic, Amenitsch and co-workers investigate the structural changes of flexible Zn-based MOF thin films upon low-pressure CO₂ uptake via the combination of quartz crystal microbalance, synchrotron radiation-based infrared spectromicroscopy and grazing incidence wide-angle X-ray scattering measurements (GIWAXS). They further shed light on triggering structural changes with the stimuli temperature and light. A particular advantage of employing these three methods is that they can be used irrespective of any topology and of the film fabrication protocol as well as substrate types. Generally, the manuscript is very-well written from both the scientific and didactic perspective, making it good to follow the workflow established for the MOF thin film characterization. Furthermore, all steps are described thoroughly and supported well by

experimental data and literature.

The idea of using MOF thin films as gas storage media as well as remotely control the uptake and release of it via e.g. light is not new and has been firstly reported by Kitagawa and co-workers on a Zn₂BDC₂DABCO-MOF incorporating azobenzene and switching the uptake and release of N₂ (see doi: 10.1021/ja2115713), which the authors also mention. The outstanding achievement in this study is the clean methodological resolution of the structural changes by combining three complementary methods that can be used independently of each other. The authors thus provide readers with a workflow that forms the basis for the analysis of flexible MOF films, their application in the investigation of gas uptake and release and thus for the development of materials that can be used technologically in the future.

I strongly recommend publication in Nature after minor revisions:

Formal issues:

- Purities of methanol and acetone are not stated, please add that accordingly
- λ etc. should be written in italic
- the authors stated they used an LED set-up in the Quartz crystal microbalance flow-cell with a wavelength of 451 nm, but within the spectra as well as the descriptions, 450 nm was used?

Scientific issues:

- The authors write "While 427 Zn₂BDC₂DABCO exhibited significant changes for Δv_{as} and Δv_{sy} (see Figure S8), only the functionalized Zn-MOF structures showed a blue-shift of the Δv_{CO_2} -ad mode with decreasing temperature, which was most pronounced for the methoxy-functionality (Table 1)." The authors should provide an explanation, why the functionalization causes this blue shift: what are the structural changes caused by the introduction of a substituent? Does this make the framework more flexible?
- The authors state "We further explored the incorporation of photo-active azobenzene molecules within the flexible Zn-MOF film pores, which offers a promising pathway to remotely trigger the CO₂ uptake and release using light as an environmentally friendly stimulus." This is not in particular true, as for azobenzene switching UV light is need for the structural E-to-Z isomerization. In this case, it is not an environmentally friendly, but an energy demanding wavelength. The authors should relativize this statement.
- The authors state that "Successful azobenzene uptake was confirmed by infrared and UV-Vis spectroscopic measurements (Figure 6, a-b), while QCM-D measurements revealed an increase in the total 485 mass related to the azobenzene uptake (Table S3)." How do they make sure that azobenzene is not adsorbed on the surface of the MOF films? This would also causes changes in the IR and UV/Vis signatures and enable photoswitching.
- In Figure 6 f it becomes apparent that the CO₂ uptake decreases with every irradiation step. The authors do not comment on this and do not explain the reason for this, which should be provided.

Reviewer #4

(Remarks to the Author)

In this work, the authors combine three operando methods to investigate the CO₂ uptake and the structural changes it engenders in a library of zinc-based metal-organic framework (MOF) films which were grown hetero-epitaxially on Cu₂BDC₂-on-Cu(OH)₂ substructures, building on earlier literature reports for the synthesis of these MOF films. The novelty of this work lies in the expert combination of grazing incidence wide-angle x-ray scattering, Fourier-transformed infrared spectroscopy, and quartz-crystal microbalance with dissipation monitoring to inform on the structure and orientation of the MOF films, their characteristic infrared fingerprints, and the CO₂ uptake triggered by either changes in temperature or light, using the conformational change of azobenzene adsorbed in the MOF films for the latter. By considering five different linkers in the Zn₂L₂DABCO MOF system, the authors could modify the degree of MOF orientation on the substrate, the structural flexibility of the MOF, and the total CO₂ uptake.

Overall, this interesting work demonstrates the strength of combining these three techniques, which may find promise beyond the promising materials studied herein. However, in its current version, the manuscript lacks mechanistic insight into what drives the preferred orientation and the structural changes upon CO₂ adsorption (*vide infra*), which seems necessary for publication in this high-profile journal.

1. From Figures 3 and S1d, it seems all MOF crystals form square rectangular prisms, the largest dimension of which is called the length L, and the smallest dimension of which is called the width W. However, the third dimension can be equal to either the length or the width; it is unclear throughout the text when one or the other possibility occurs. Furthermore, Figure 2 reports the fraction of crystals with oriented growth which, if I understand correctly, correspond to crystals whose long dimension (the length) falls perpendicular to the plane of the MOF/substrate interface (= the MOF film thickness). However, the SEM micrographs in Figure 3 show a strong variation in both the crystal dimensions and the orientation, even within a given MOF system. As a result, only providing the fraction of crystals that show oriented growth seems insufficient, especially when trying to understand the resulting flexibility of the material upon CO₂ adsorption. Hence, it would tremendously help the discussion if the authors could provide, for the four MOF films that show growth, a distribution of the crystal cell dimensions and the angle between the shortest dimension of the crystal and this interfacial plane, as the latter directly relates to the oriented growth.

2. Figure 3f is confusing. First, the Cu₂BDC₂ substructure shows BDC linkers surrounding the Cu₂ paddlewheel units in all three directions despite being a 2D MOF. The substructure shown in this schematic is Cu₂BDC₃ instead, with an extra BDC linker binding to the axial Cu₂ paddlewheel positions. If this were correct, how would this binding take place? Second, the b direction in Cu₂BDC₂ is about half that of the Zn-MOFs (see Table S1), so every other Cu₂BDC₂ layer should remain disconnected from the Zn-MOF. Finally, the DABCO ligand should connect to the axial positions in the MOF; the four

paddlewheel positions being occupied by BDC linkers.

3. Can the authors relate the observed frequency shifts upon CO₂ or azobenzene adsorption to changes in cell parameters and, hence, flexibility? Do any events induce a partial transition from the LP to the CP phase? Can they quantify the statement on page 23 reading “This difference in behaviour is attributed to an increase in structural flexibility owed to introduced linker functionalities”?

4. Can the authors report on the location and distribution of the adsorbed CO₂ molecules both within the MOF film and within a MOF unit cell? A similar question holds for azobenzene since the loading level indicates only a fraction of the unit cell is occupied. Do azobenzene-loaded unit cells conglomerate? Where are the 13% of the azobenzene molecules that isomerise located?

5. Did the authors carry out GIWAXS experiments after the CO₂ has been evacuated from the MOF in Figure 5? This is necessary to distinguish between reversible and irreversible changes in the MOF.

6. What causes the distinct drop in CO₂ uptake both during CO₂ loading and upon switching in Figure 6e,f? Did the authors check the cyclability of temperature-induced CO₂ adsorption?

7. The authors denote the absence of a band around 1690 cm⁻¹ to the full conversion of -COOH groups to -COO- groups. However, for the Cu₂BDC₂ substrate, shouldn't these -COOH groups still be present at the surface where the Zn-MOF will grow in a subsequent step?

8. When discussing the fabrication of the pristine Zn-MOF films, no source of zinc ions is present when immersed in the linker-based solutions (lines 152-157). How does the MOF growth then take place?

9. Why was the nonfunctionalised Zn₂BDC₂DABCO not considered for azobenzene loading?

10. Lines 150-151 mention “methanolic” and “EtOH” simultaneously.

11. Is there a reason that a Delta symbol precedes the frequencies in Figure 4a, b, and h, since they show the actual frequencies and no frequency shifts?

Version 1:

Reviewer comments:

Reviewer #1

(Remarks to the Author)

We are happy with the response from the authors. One minor issue is that the SEM in Figure 3D is significantly charged, which makes it difficult to clearly support the claim in the context.

Reviewer #4

(Remarks to the Author)

I thank the authors for their clarifications and revisions in the manuscript, which cover the initial reservations I had. As a result, I'm happy to suggest this manuscript for publication in Nature Communications.

As a final suggestion, I would encourage the authors to include in their final discussion not only the strength of combining the three reported techniques but also the main challenges that remain unresolved, for instance, how to probe the distribution of the adsorbed CO₂ molecules in the MOF film.

Subject: Major revisions of manuscript NCOMMS-24-65363

Please find enclosed our manuscript entitled “*Flexible metal-organic framework films for reversible low-pressure carbon capture and release*”. We are thankful to all the reviewers for their insightful suggestions as well as for having received this opportunity to resubmit our revised manuscript for which we have addressed carefully all the reviewers’ comments. Any changes made during the revision process are highlighted in yellow in the main manuscript and the supplementary information, and our point-by-point responses to the comments are given below.

Responses to Reviewer #1: The current work presents the synthesis of flexible MOF films for low pressure CO₂ adsorption, utilizing three operando methods to study the structural changes of MOFs and their CO₂ adsorption/desorption capacity. The structural changes are induced by CO₂ adsorption or stimuli such as temperature and light. Due to the lack of novelty in the synthesis and characterization, and the insufficient explanation of the CO₂ adsorption/desorption process and structural change mechanism, I do not recommend this paper for publication in Nature Communications. Below are some questions and suggestions for the authors to address.

Comment: The role of DABCO is unclear. Based on Figure 2 and 3, DABCO acts as free ligands and is not integrated into MOF framework. The role of DABCO needs to be discussed.

Response: We thank the Reviewer for the insightful question. To provide a detailed explanation of the role of the DABCO linker, we begin by addressing its integration in the bulk {Zn₂(1,4-bdc)₂dabco} system. Specifically, we clarify that DABCO is incorporated into the {Zn₂(1,4-bdc)₂dabco} structure, as demonstrated by single-crystal structures reported in the literature. The most relevant references are cited in our manuscript, and we summarize their findings below.

The preparation protocol of {Zn₂(1,4-bdc)₂dabco} was first reported by Dybsteve et al. (*Dybtsev, D. N., Chun, H. & Kim, K., Angewandte Chemie (International ed. in English) 43, 5033–5036 (2004)*), alongside its physico-chemical and structural characteristics. The guest-free structure consists of dinuclear Zn₂ units arranged in a paddle-wheel configuration, bridged by 1,4-terephthalates acting as dianions thus forming a distorted two-dimensional {Zn₂(1,4-bdc₂)} square grid. Along the axial positions of the paddle wheels, DABCO links the {Zn₂(1,4-bdc₂)} square grids acting as a pillaring linker, often also termed as a neutral donor pillar (*Senkovska, I. et al. Angewandte Chemie (International ed. in English) 62, e202218076; (2023)*), constituting to the three-dimensional tetragonal structure (*P4/mmm*), with the single crystal file reported under the number CCDC-238860 (*Dybtsev, D. N., Chun, H. & Kim, K., Angewandte Chemie (International ed. in English) 43, 5033–5036 (2004)*). In contrast, MOF-5 composed only of {Zn₂(1,4-bdc₂)}, crystallizes in a cubic space group with a significantly lower surface area compared to the pillared {Zn₂(1,4-bdc)₂dabco} (*Eddaoudi, M., Jaheon, K., Rosi, N., Vodak, D., Wachter, J., O’Keeffe, M., Yaghi, O., Science, 295, 469-472 (2002)*). Therefore, the DABCO ligand plays a crucial role in defining the {Zn₂(1,4-bdc)₂dabco} framework structure. This holds also true for chemically functionalized Zn-MOF structures, bearing single, multiple (*Xie, M., Prasetya, N., Ladewig, B. P., Inorganic Chemistry Communications 108, 107512; (2019)*; *Henke, S., Schneemann, A., Wütscher, A. & Fischer, R. A. Journal of the American Chemical Society 134, 9464–9474; 10.1021/ja302991b (2012)*.) or combinations of functionalized groups in the terephthalate linker (*Hahm, H., Yoo, K., Ha, H., Kim, M. Inorganic chemistry 55, 7576–7581; (2016)*).

Upon incorporation of azobenzene in the $\{Zn_2(1,4\text{-bdc})_2\text{dabco}\}$ pores, the volume per dinuclear Zn_2 unit decreases by 127.3 \AA^3 accompanied by a change in the crystal system from tetragonal to orthorhombic (*Cmmm*), as reported for benzene as a guest molecule (N. Yanai, T. Uemura, M. Inoue, R. Matsuda, T. Fukushima, M. Tsujimoto, S. Isoda and S. Kitagawa, *J. Am. Chem. Soc.*, **134**, 4501–4504 (2012); CCDC-238861). Upon photo-irradiation of the guest-host composite, the structure transitions between the tetragonal and orthorhombic systems, a process demonstrated Yanai *et al.* using electron-beam diffraction on single particles. Notably, the $\{Zn_2(1,4\text{-bdc})_2\text{dabco}\}$ system retains the pillaring DABCO molecule even when exposed to significant pressures as simulations of Wieme *et al.* have shown (Wieme, J., Rogge, S. M. J., Yot, P. G., Vanduyfhuys, L., Lee, S.-K., Chang, J.-S., Waroquier, M., Maurin, G. and van Speybroeck, V., *J. Mater. Chem. A*, **7**, 22663 (2019)).

In the case of the heteroepitaxial $\{Zn_2(1,4\text{-bdc})_2\text{dabco}\}$ film, also reported in the present manuscript, the growth of the $\{Zn_2(1,4\text{-bdc})_2\text{dabco}\}$ structure has been described in detail also in some of our earlier works (Klokic, S., Naumenko, D., Marmiroli, B., Carraro, F., Linares-Moreau, M., Zilio, S. D., Birarda, G., Kargl, R., Falcaro P., and Amenitsch, H., *Chem. Sci.*, **13**, 11869–11877 (2022); Klokic, S., Marmiroli, B., Naumenko, D., Birarda, G., Zilio, S. D., Velasquez-Hernández, M. J., Falcaro P., Vaccari, L., Amenitsch, H., *CrystEngComm* **26**, 2228–2232; (2024).). The Zn-MOF structure show a lattice matching with the lower Cu_2BDC_2 layer, facilitating heteroepitaxial film growth. The resulting Zn-MOF structure accommodates azobenzene in its three-dimensional pores. GIWAXS patterns and infrared spectroscopy confirmed the *P4/mmm* crystal system and the heteroepitaxial nature of the films. The findings reported here for the isostructural functionalized MOF films align with these results. In addition, QCM-D measurements show no CO_2 adsorption in the case of the pure Cu_2BDC_2 layer (Figure S6, a), whilst the pillared Zn-MOF structures show very pronounced sorption characteristics (Figure S6, b-d).

We would like to highlight that the following references related to the DMOF-1 crystal structure and its structural properties are cited in our manuscript:

Reference 39: Henke, S., Schneemann, A., Wütscher, A. & Fischer, R. A. Directing the breathing behavior of pillared-layered metal-organic frameworks via a systematic library of functionalized linkers bearing flexible substituents. *Journal of the American Chemical Society* **134**, 9464–9474; 10.1021/ja302991b (2012).

Reference 43: Yanai, N. *et al.* Guest-to-host transmission of structural changes for stimuli-responsive adsorption property. *Journal of the American Chemical Society* **134**, 4501–4504; 10.1021/ja2115713 (2012).

Reference 50: Xie, M., Prasetya, N. & Ladewig, B. P. Systematic screening of DMOF-1 with NH_2 , NO_2 , Br and azobenzene functionalities for elucidation of carbon dioxide and nitrogen separation properties. *Inorganic Chemistry Communications* **108**, 107512; 10.1016/j.inoche.2019.107512 (2019).

Reference 65: Hahm, H., Yoo, K., Ha, H. & Kim, M. Aromatic Substituent Effects on the Flexibility of Metal-Organic Frameworks. *Inorganic chemistry* **55**, 7576–7581; 10.1021/acs.inorgchem.6b00983 (2016).

Reference S1: Dybtsev, D. N., Chun, H. & Kim, K. Rigid and flexible: a highly porous metal-organic framework with unusual guest-dependent dynamic behavior. *Angewandte Chemie (International ed. in English)* **43**, 5033–5036; 10.1002/anie.200460712 (2004).

Reference S7: Senkovska, I. *et al.* Understanding MOF Flexibility: An Analysis Focused on Pillared Layer MOFs as a Model System. *Angewandte Chemie (International ed. in English)* **62**, e202218076; 10.1002/anie.202218076 (2023).

Comment: There is insufficient information on how structural changes occur. Detailed descriptions of the accurate structures before and after the structural changes (e.g. degree of ligand rotation) are necessary.

Response: We thank the Reviewer for raising concerns about the structural changes in the isorecticular Zn-MOF films reported in this manuscript. Below, we provide a more comprehensive summary of our results highlighting the information related to the description of the structures before and after structural changes, along with the results related to the structural changes.

1. Before structural changes: A detailed description and characterization of the Zn_2L_2DABCO structures were conducted using various techniques:
 - *GIWAXS measurements:* By in-plane and out-of-plane analyses (Figure S4), we assessed the lattice parameters (Table S1), the lattice mismatch relative to the underlying Cu_2BDC_2 -on- $Cu(OH)_2$ structure (Table S2), and the degree of lattice orientation (Figure S5). Additionally, the crystallite alignment was evaluated by an orientation analysis of the GIWAXS pattern, and the results were confirmed by SEM micrographs (Figure S1). The Zn_2L_2DABCO structures with $L = BDC, Me-BDC$ and $MeO-BDC$ grow isostructural.
 - *Azimuthal angle dependence measurements* were conducted in the scope of the reviewing process to corroborate the epitaxial alignment of the Zn_2L_2DABCO structures with respect to the Cu_2BDC_2 sublayer. The respective data is summarized in the Supplementary Information in chapter 6 with Figure S4 being adapted as well as in the main manuscript.
 - *IR-spectromicroscopy measurements* confirmed the isostructural growth of the functionalized structures (Figure 4, CO_2 OFF).

The isorecticular growth of the functionalized structures with respect to Zn_2BDC_2DABCO is discussed in the main manuscript in *lines 289 – 311*.

2. After structural changes: Characterization of Zn_2L_2DABCO after structural changes using the following approaches to verify CO_2 ad- and desorption, and its influence on structural flexibility due to varying BDC-linker functionalities:
 - *At 295 K after CO_2 uptake,* the Zn-MOF structures were characterized by IR-spectromicroscopy, GIWAXS and QCM-D measurements. Here a contraction/expansion of the Zn_2L_2DABCO pores was confirmed by GIWAXS results demonstrating a deformation occurring either in the out-of-plane ($Zn_2Me-BDC_2DABCO$), or both in the out-of-plane and in-plane directions ($Zn_2MeO-BDC_2DABCO$ and Zn_2BDC_2DABCO , see Figure 5). This process is caused by the adsorption of CO_2 as evidenced by the adsorption capacities evaluated through QCM-D measurements (Table S3) that are further confirmed by the appearance of the characteristic vibrational mode related to adsorbed CO_2 in IR-spectromicroscopy measurements (Figure 4). These results show that during CO_2 sorption the Zn-MOF film structures expand in the direction parallel to the substrate, whilst only the most flexible methoxy system shrinks in the in-plane direction.
 - *At 295 K, 240 K and 200 K:* Low-temperature studies were conducted to prove the flexibility of the Zn-MOF films where a large pore to narrow pore (*LP-to-NP*) transition was demonstrated during IR-spectromicroscopy measurements. Here, in the first step this transition was conducted in the absence of CO_2 , and subsequently, in the presence of CO_2 molecules. This *LP-to-NP* caused a change in the MOF pore (contraction) which was translated to the CO_2 molecules, thus confirming their successful adsorption within the MOF pores (Figure S7-8, Table 1). The methoxy-functionalized structure was found to be the most flexible in the series.

- Upon azobenzene uptake, a strong contraction for the functionalized Zn₂L₂DABCO structures (L = Me-BDC, MeO-BDC) was observed with the evaluation of the changes in d-spacing provided in Figure S13. Due to the pore contractions, the CO₂ sorption capacity was found to shrink for all Zn₂L₂DABCO structures with the results provided in Table S3. Moreover, the Zn-MOF films were shown by IR-spectromicroscopy and QCM-D measurements to respond to the isomerization of the chromophore when triggered by light. In the case of Zn₂BDC₂DABCO, earlier polarization-dependent experiments showed that the transition dipole moment of azobenzene aligns along the *c*-axis direction (Klokic, S. et al. *Chemical science* 13, 11869–11877; 10.1039/d2sc02405e (2022)). The structural transition orients along the DABCO linker which is typically evidenced by changes in the (001) reflection and modulations of the IR vibrational mode at ~770 cm⁻¹ as shown in the insets in Figure 6c-d.

The process of the structural changes after CO₂ uptake are given in several parts of the manuscript, with the most important in *lines 434 – 450, lines 476 – 490 and lines 530 – 544.*

We are further thankful for the suggestion that exemplarily the degree of ligand rotation can be assessed to represent structural changes, which is commonly done by magic angle spinning (MAS) in solid-state NMR spectroscopy on bulk MOF structures (Burch, N. C., et al.; *J. Phys. Chem. Lett.* 6, 812–816 (2015)). Recently, there have been efforts in developing solid-state NMR for thin films, where most of the studies performed use free-standing films, nanosheets or deposited on membranes (Nokab, M. H., Sebakhy, K. O., *Nanomaterials*, 11 (6):1494 (2021)). To the best of our knowledge, in the case of inorganic-organic composites the NMR active nucleus that could be probed is ¹H, however, to allow a statement on structural changes within the Zn-MOF film, ^{14,15}N would be more meaningful as this nucleus exists only in the Zn-structure. According to the Review of Nokab et al., there is no suitable solid-state NMR technique available for this kind of chemical-connectivity. Moreover, for thin films, sensitivity is troublesome as the measured signal and broadness of the overall spectrum is related to the active nuclear spins.

As noted by Svenkovska et al. (Senkovska, I. et al. *Angewandte Chemie (International ed. in English)* 62, e202218076 (2023)), these thermal motions of framework constituents are rather local mobility effects and are not necessarily related to the overall structural transitions observed for flexible MOFs, *i.e.*, LP to narrow-pore NP pore size transition. The latter is what we term as “structural changes” in the present manuscript (*line 408 – 420*), referring to global changes in the crystal lattice. In other words, how the Zn₂L₂DABCO structure changes in the presence of CO₂ molecules when triggered by stimuli including temperature and light (*line 96 – 102*).

Comment: *The authors did not explain why reversible structural changes occur in some MOFs but not in others following CO₂ capture and release.*

Response: This aspect is not explicitly discussed in the present manuscript, as our primary focus is on low-pressure CO₂ adsorption and the methodological approaches used to assess it, exemplified by the flexible Zn₂L₂DABCO structures. We thank the Reviewer for the inquiry, and we would like to elaborate further.

In general, structural changes in MOFs occur in various forms. In this manuscript, we primarily refer to structural deformations which term as “flexibility” or “switchability”. These deformations include gating (one-step transformation) or breathing (two-step transformation), where the latter is introduced in the main manuscript (*lines 96 – 99*).

In the case of pillared layer frameworks such as the Zn_2L_2DABCO , the breathing process has been extensively reported in the presence of various guest molecules. These molecules can induce a reversible transition between a guest-filled open-pore phase (OP), or, in the case of Zn_2BDC_2DABCO , a large-pore phase (LP), and a guest-free narrow-pore phase (NP). The structural flexibility of this MOF class of pillared layer frameworks arises from its unique assembly, where 2D $\{Zn_2(1,4-bdc_2)\}$ layers are connected by DABCO pillars to form a 3D-framework as reported by *Buraekaw, S. et al. (Journal of Materials Chemistry, 22, 10249 (2012))*. For this class of flexible MOFs (third generation), such structural transitions can be triggered either by external stimuli (*i.e.*, temperature, light) or by the adsorption of molecules which act pore-opening (*i.e.*, CO_2). Other MOFs exhibiting similar flexibility include DUT-8 and DUT-8(Ni), DUT-131, DUT-128, MIL-53. A comprehensive overview of MOF flexibility is provided in the excellent review of *Senkovska, I. et al. Angewandte Chemie (International ed. in English) 62, e202218076 (2023)*.

Finally, we would like to highlight the section in the main manuscript (*lines 408 – 412*) where the reasons for structural transitions in Zn-MOFs, specifically the LP-to-NP transition, are discussed:

To investigate interactions between adsorbed CO_2 and the Zn-MOF host structure, we leveraged the flexibility of the frameworks. Upon guest molecule uptake, Zn_2BDC_2DABCO and similar isostructural analogues^{39,50,59,60} undergo a structural transformation, owed to their pillared-layered assembly, in which bulk systems reversibly transform between a large-pore (LP) and narrow-pore phase (NP).⁴¹ This process commences upon gas ad- or desorption and is often termed “breathing” because of the reversible expansion or contraction of the pores.^{61,62}

Comment: Cyclability tests for CO_2 adsorption/desorption are needed.

Response: We thank the Reviewer for addressing this point as it is not clearly elaborated in the present manuscript. Considering that one cycle refers to CO_2 adsorption followed by desorption, we would like to emphasize that the CO_2 sorption measurements presented here were conducted as cycling measurements. For all the three techniques described, the initial films were purged extensively by nitrogen, followed by sequential CO_2 adsorption (CO_2 ON) and desorption (CO_2 OFF) steps. This process was repeated for a total of two (GIWAXS), three (infrared spectromicroscopy and QCM) or more cycles (QCM, *see Fig. 6f*). Specifically, for infrared spectromicroscopy, three cycles were performed for every MOF film at 295 K and as duplicates because of time reasons for the lower temperatures (240 K, 200 K), where five spatially different points on the sample were measured to provide a more representative response of the film.

To clarify this point we have included in the section “Methods” the following additions as highlighted in yellow:

For every film, duplicate measurements were performed at an acquisition time of 60 s and an incident grazing angle of $\alpha_i = 0.2^\circ$. (line 182 – 183)

This procedure was repeated at 295 as triplicates, at 240 K and 200 K for time reasons as duplicates. (line 207)

Comment: The authors attributed the low CO_2 adsorption capacity in Zn-MOF/AB to structural contraction. However, pore blockage by AB can significantly effect CO_2 capture. This part should be discussed in detail.

Response: We are thankful to the Reviewer for the concerns raised, and we agree that pore blockage by azobenzene is certainly a factor influencing CO_2 adsorption capacity. To clarify our line of argumentation, we would like to summarize more concisely our findings. We certainly do observe a

pronounced structural shrinkage as shown by GIWAXS results, which is only related to the azobenzene uptake itself (Figure S12, see main manuscript *lines 515 – 521*). In the presence of azobenzene within the Zn-MOF pores, the decreased quantity of the CO₂ adsorption capacity is shown in Figure S14 and summarized in more detail in Table S3, as described in the Supplementary Information (section 8):

After infiltrating the Zn-MOF films by azobenzene, a decrease in CO₂ adsorption capacity was observed, with the decreased quantity being influenced by the number of azobenzene molecules within the pores (see Table S4 for AB loading level).

We quantified an azobenzene loading level of 0.7 for the methyl- and 0.2 for the methoxy-functionalized Zn-MOF/AB structures. These two values differ, indicating that pore blockage by azobenzene would vary and one would expect that the quantity of CO₂ uptake would be higher in the case of the methoxy-functionalized Zn-MOF/AB than in the methyl-structure. However, QCM results clearly show that the adsorbed CO₂ quantity is equal irrespective of the azobenzene loading level ($0.04 \pm 0.01 \mu\text{g}/\text{cm}^2$, Figure 6e). Hence at a lower azobenzene loading in the case of Zn₂MeO-BDC₂DABCO/AB and thus at a lower azobenzene pore blockage, the CO₂ uptake remains equal to that where a higher pore blockage, i.e., azobenzene loading, is present (Zn₂Me-BDC₂DABCO/AB). Because of this result, we are inclined to attribute the decrease in CO₂ adsorption mainly to the structural change of the MOF pore caused by accommodating the azobenzene molecules. The pore contraction is expected to follow a large-pore (no azobenzene) to narrow-pore transition (with azobenzene) (see exemplarily references 41 and 64 in main manuscript, or Schaper L., Keupp J., Schmid R., *Front. Chem.* 9, 757680 (2021)). To clarify this point raised by the Reviewer, we have modified the respective lines 525 – 528:

QCM-D results show a significant decrease in CO₂ uptake (Figure S14, a-b) with the adsorbed amount in the functionalized Zn-MOF/AB accounting for $0.04 \pm 0.01 \mu\text{g}/\text{cm}^2$ (Figure 6e). These data show that the azobenzene molecules still leave sufficient space inside the MOF for CO₂ uptake²⁹ considering the respective azobenzene loading levels.

We would also like to highlight that the error in referencing was corrected in line 520:

Here, Zn₂MeO-BDC₂DABCO/AB experiences a pronounced structural shrinkage, with $\Delta d = 0.22 \text{ \AA}$ for the (100) reflection (Figure S13, b), compared to $\Delta d = 0.08 \text{ \AA}$ for Zn₂Me-BDC₂DABCO/AB.

Comment: The CO₂ sorption capacity of AB functionalized MOF significantly decreased after 4 cycles, particularly for Zn₂MeO. The authors should provide explanations.

Response: We kindly thank the Reviewer for the remark and have taken the liberty to include two additional graphs in the Supplementary Information to explain our findings better (Figure S12 c-d): The decline in sorption capacity is mainly related to the decrease in the photo-switching fraction of *trans-to-cis* azobenzene. In the case of Zn₂Me-BDC₂DABCO, when comparing the first and second photo-switch, IR results show only a decrease by 2% (Figure S12 c). In contrast, for Zn₂MeO-BDC₂DABCO, this decrease is significantly higher with 11% (Figure S12 d). This decrease is attributed to the functional groups present in the Zn-MOF structure, as this behaviour was not found in the non-functionalized system reported previously by some of the present authors (Klokic, S. *et al. Chemical science* 13, 11869–11877, (2022)).

To clarify this point raised by the Reviewer, we have included two additional graphs in Figure S12 c-d:

Figure S12 (...) (c) $Zn_2Me-BDC_2DABCO$ shows 21% of azobenzene photo-switching after the first cycle and 19% after the second. (d) $Zn_2MeO-BDC_2DABCO$ shows 22% of azobenzene photo-switching after the first cycle and 9% after the second. This decrease is attributed to the functional groups present in the Zn-MOF structure, as this behaviour is not found in the non-functionalized system.²

And we have included an explanation in the main manuscript in line 543 – 544:

The decrease in sorption capacity after the first photo-switch is related to the lowering of the trans-to-cis isomerization fraction (Figure S12, c-d).

Comment: The adsorption capacity (mmol/g) of AB functionalized MOFs should be provided for comparison.

Response: We kindly thank the Reviewer for the remark. However, we would like to note that the adsorption capacities were already included in the Supplementary Table S3 as Entry 5.

Comment: In figure 6e and S14, the uptake difference from QCM-D is below 0, with some values remaining negative even after reaching the plateau. Since this is closely related to equipment accuracy, the authors should explain.

Response: We kindly thank the Reviewer for the remark. We agree that in some cases the uptake difference from QCM-D is below zero or negative after purging the CO_2 exposed layers with N_2 . However, we do not think that the method's accuracy is the reason for this observation. The QCM-D has a mass sensitivity of 0.5 ng/cm^2 and an LOD of 1.34 ng/cm^2 and a drift of 1 Hz/h ($17.7 \text{ ng/cm}^2 \text{ h}$), according to the manufacturer.

In the case of Figure 6e, the QCM-D experiments were repeated 4 times and a STD for the films was achieved of $Zn_2MeO-BDC_2DABCO/AB$ $0.03 \pm 0.01 \text{ } \mu\text{g/cm}^2$, $Zn_2Me-BDC_2DABCO$ $0.09 \pm 0.05 \text{ } \mu\text{g/cm}^2$, Zn_2BDC_2DABCO $0.04 \pm 0.01 \text{ } \mu\text{g/cm}^2$.

We rather suggest that N_2 causes the desorption of residual solvents or other volatiles (*i.e.*, MeOH, EtOH) from the film, which was clearly seen during the preconditioning of the samples (see graph below, A). Because of this we purged the films and the sample compartment thoroughly to ensure a stable baseline. Also, N_2 itself is desorbed during CO_2 exposure, which explains the prolonged decrease of the signal in Figure 6e and S14. Upon N_2 exposure afterwards, CO_2 desorbs fully as evidenced by IR-spectromicroscopy measurements (see graph below, B), but the changed film structure does not allow N_2 or (purge volatiles) to fill the same vacancies as before, causing a zero to negative net signal. The following graphs were further included in Figure S6:

As closely related questions to this topic have been risen by Reviewer #1 and #4, we have included the explanation to this behaviour in the Supplementary Information in Figure S6, legend:

The transient response of the signal likely reflects pressure changes when switching between N₂ (CO₂ OFF) and CO₂ (CO₂ ON), affecting sensor oscillation.

We have further included the information on the CO₂ adsorption capacities for the Zn-MOF after infiltration by azobenzene in the Supplementary Information in Figure S14, legend:

The QCM-D experiments were repeated 4 times and a STD for the films was achieved of Zn₂MeO-BDC₂DABCO/AB $0.03 \pm 0.01 \mu\text{g}/\text{cm}^2$, Zn₂Me-BDC₂DABCO $0.09 \pm 0.05 \mu\text{g}/\text{cm}^2$, Zn₂BDC₂DABCO $0.04 \pm 0.01 \mu\text{g}/\text{cm}^2$.

In the case of Zn-MOF films, the comparably much stronger increase of the response signal is probably also related to the swelling of the layer due to CO₂ adsorption followed by its stabilization of this new configuration. This is typically observed in QCM-D measurements, as exemplarily given in reference S12. We have included this explanation in chapter S8 in the Supplementary Information:

In the case of the Zn-MOF films the type of response shown in Figure S6 indicates that the only the MOF layer is highly sensitive towards CO₂. This is evidenced as a rapid increase in the response occurs due to the adsorption of the analyte caused by swelling of the Zn-MOF film and followed by a drop in response until the stabilization of the CO₂ adsorbed layer.¹²

And we have referenced this aspect also in the main manuscript in line 387 – 388:

Results showed an immediate response of the Zn-MOF films towards CO₂ with strong and rapid gas adsorption, much in contrast to the responses obtained for the bare crystal or Cu₂BDC₂ substructure (see Supplementary Information chapter 8 and Figure S6, a).

Responses to Reviewer #2: *I co-reviewed this manuscript with one of the reviewers who provided the listed reports. This is part of the Nature Communications initiative to facilitate training in peer review and to provide appropriate recognition for Early Career Researchers who co-review manuscripts.*

Response: We kindly thank the Reviewer for co-reviewing our manuscript.

Responses to Reviewer #3: In their manuscript, Klokic, Amenitsch and co-workers investigate the structural changes of flexible Zn-based MOF thin films upon low-pressure CO₂ uptake via the combination of quartz crystal microbalance, synchrotron radiation-based infrared spectromicroscopy and grazing incidence wide-angle X-ray scattering measurements (GIWAXS). They further shed light on triggering structural changes with the stimuli temperature and light. A particular advantage of employing these three methods is that they can be used irrespective of any topology and of the film fabrication protocol as well as substrate types. Generally, the manuscript is very-well written from both the scientific and didactic perspective, making it good to follow the workflow established for the MOF thin film characterization. Furthermore, all steps are described thoroughly and supported well by experimental data and literature.

The idea of using MOF thin films as gas storage media as well as remotely control the uptake and release of it via e.g. light is not new and has been firstly reported by Kitagawa and co-workers on a Zn₂BDC₂DABCO-MOF incorporating azobenzene and switching the uptake and release of N₂ (see doi: 10.1021/ja2115713), which the authors also mention. The outstanding achievement in this study is the clean methodological resolution of the structural changes by combining three complementary methods that can be used independently of each other. The authors thus provide readers with a workflow that forms the basis for the analysis of flexible MOF films, their application in the investigation of gas uptake and release and thus for the development of materials that can be used technologically in the future.

I strongly recommend publication in Nature after minor revisions:

Formal issues:

Comment: Purities of methanol and acetone are not stated, please add that accordingly

Response: We thank the Reviewer and have included the purities in lines 140 and 141 in the main manuscript accordingly.

Methanol (MeOH, 99.8%) and absolute ethanol (EtOH, 99.8%) were bought from VWR Chemicals, while acetone was purchased from AvantorTM (99.8%).

Comment: λ etc. should be written in italic

Response: We thank the Reviewer and have modified all Greek letters accordingly with the changed given below:

Line 169: (...), where q denotes the length of the scattering vector ($q = \frac{4\pi}{\lambda} \sin\left(\frac{2\theta}{2}\right)$), λ being the wavelength (0.154 nm, 8 keV) and θ the scattering angle.

Comment: the authors stated they used an LED set-up in the Quartz crystal microbalance flow-cell with a wavelength of 451 nm, but within the spectra as well as the descriptions, 450 nm was used?

Response: We are grateful to the Reviewer for pointing out this error. The used LED (laser diode, metal can) has a peak wavelength of 450 nm (± 10 nm). The line 220 has been corrected accordingly:

On top of the cell a LED setup was glued tightly, equipped with a blue LED (450 nm, Osram, Germany) and a UV LED (365 nm, Würth, Germany).

Scientific issues:

The authors write "While 427 Zn₂BDC₂DABCO exhibited significant changes for Δvas and Δvsy

(see Figure S8), only the functionalized Zn-MOF structures showed a blue-shift of the $\Delta\nu_{\text{CO}_2\text{-ad}}$ mode with decreasing temperature, which was most pronounced for the methoxy-functionality (Table 1).“ The authors should provide an explanation, why the functionalization causes this blue shift: what are the structural changes caused by the introduction of a substituent? Does this make the framework more flexible?

Response: We are grateful for the questions risen by the reviewer. The introduction of substituents (methoxy, methyl) caused an increase of the Zn-MOF unit cell (1 – 2% with respect to $\text{Zn}_2\text{BDC}_2\text{DABCO}$, Table S1), which was also observed in bulk Zn-MOF structures of increasing ligand sizes as reported by Henke et al. (reference 39, main manuscript).

The methoxy-functionality makes the Zn-MOF framework significantly more flexible compared to the methyl or the non-functionalized structure. This increase in structural flexibility due to linker functionalization is best evidenced in the case of azobenzene infiltration, where a significantly stronger contraction of the methoxy structure was observed when compared to the methyl one (see ESI, Figure S13). These differences in structural flexibility are also evidenced during CO_2 sorption experiments using GIWAXS where an expansion in out-of-plane and in-plane direction was again observed for the methoxy structure, whilst the methyl showed only out-of-plane changes (see lines 476 – 482). This behavior is further reciprocated by infrared spectromicroscopy results, where the changes of the IR band shifts for the low-temperature study were attributed to structural flexibility (lines 432 – 433).

Line 305 – 308: Because of this lattice orientation for both $\text{Zn}_2\text{MeO-BDC}_2\text{DABCO}$ and $\text{Zn}_2\text{Me-BDC}_2\text{DABCO}$ with respect to the lower Cu_2BDC_2 structure, both systems result in being more flexible compared to the $\text{Zn}_2\text{BDC}_2\text{DABCO}$ system (see ESI for more details in chapter 6 and Figure S4, g).

Line 432: These differences are attributed to enhanced structural flexibility that is supported by QCM results showing a 30% higher CO_2 uptake for functionalized Zn-MOF films.

Lines 487 – 490: This difference in behaviour is attributed to an increase in structural flexibility owed to introduced linker functionalities,^{51,65} which further increase the amount of adsorbed CO_2 stems due to interactions arising between the adsorbate and the functionalized Zn-MOF structure.

As highlighted by the Reviewer, the blue-shift of the $\Delta\nu_{\text{CO}_2\text{-ad}}$ mode with decreasing temperature is not clearly attributed to the differences in framework flexibility, for which we modified the respective passage accordingly:

Lines 441 – 444: These findings strongly support the increase in structural flexibility caused by the linker functionalization as well as the presence of adsorbed CO_2 molecules within the MOF lattice, which interact with the functional groups of the BDC linkers.

Within the scope of the reviewing process, we have further performed azimuthal angle dependence measurements on the $\text{Zn}_2\text{L}_2\text{DABCO}$ structures. These results are now included in Figure S4, from which we concluded that also the alignment of the upper Zn-MOF structure with respect to the lower Cu_2BDC_2 system might be responsible for the structural flexibility observed for the methoxy and methyl functionalized systems. Both the $\text{Zn}_2\text{MeO-BDC}_2\text{DABCO}$ and the $\text{Zn}_2\text{Me-BDC}_2\text{DABCO}$ systems align with their *a*-axis being parallel to the *a*-axis of the Cu_2BDC_2 structure (as $(100)_{\text{Zn}_2\text{L}_2\text{DABCO}}$ and $(100)_{\text{Cu}_2\text{BDC}_2}$ are coinciding in φ -scan), and orthogonally to this, their *c*-axes match the Cu_2BDC_2 *b*-axis (as $(001)_{\text{Zn}_2\text{L}_2\text{DABCO}}$ is 90° shifted with respect to $(100)_{\text{Cu}_2\text{BDC}_2}$ in φ -scan). The resulting lattice alignment is schematically depicted in the Supplementary Information in Figure S4 (also shown below). The $\text{Zn}_2\text{BDC}_2\text{DABCO}$ structure shows this alignment, however it comprises also a 90° rotated lattice matching in which the *a*-axis matches the *b*-axis of the Cu_2BDC_2 system and orthogonally to it the *c*-

axis aligns with the *a*-axis of the sublayer. This rotation in alignment can explain the observed differences in structural flexibility: The *c*-axis of the Zn-MOF bears no covalently interlinked components since only the pillaring DABCO molecule bridges between the zinc paddlewheel units. Similarly, the *b*-axis of Cu₂BDC₂ bears no covalent interlinking amongst the Cu₂BDC₂ sheets and is often referred as to provide more flexibility to the MOF structure (*Falcaro et al., Nature materials, 16(3), 342-348 (2017)*). If both rather flexible axes are coinciding parallel to each other, this direction of the lattice allows considerably higher flexibility, which can also explain the success of this alignment despite the rather high mismatch ratio (see Table S2). In contrast, if the *a*-axis of the Zn-MOF, which comprises covalently interlinked terephthalate units with zinc-paddlewheel moieties, aligns onto the *c*-axis of Cu₂BDC₂, the flexibility of the entire film system is compromised. This alignment allows both the Zn₂MeO-BDC₂DABCO and the Zn₂Me-BDC₂DABCO structures to have a higher structural flexibility, as it is seen also in the CO₂ sorption measurements.

Comment: The authors state “We further explored the incorporation of photo-active azobenzene molecules within the flexible Zn-MOF film pores, which offers a promising pathway to remotely trigger the CO₂ uptake and release using light as an environmentally friendly stimulus.” This is not in particular true, as for azobenzene switching UV light is need for the structural E-to-Z isomerization. In this case, it is not an environmentally friendly, but an energy demanding wavelength. The authors should relativize this statement.

Response: We are thankful to the Reviewer for pointing out this agreeably too exaggerating choice of words and we have relativized the sentence in question accordingly:

Lines 491 – 494: *We further explored the incorporation of photo-active azobenzene molecules within the flexible Zn-MOF film pores, which may provide a promising pathway for remotely influencing CO₂ uptake and release using LED light as an energetically efficient stimulus.*

Comment: The authors state that “Successful azobenzene uptake was confirmed by infrared and UV-Vis spectroscopic measurements (Figure 6, a-b), while QCM-D measurements revealed an increase in the total 485 mass related to the azobenzene uptake (Table S3).” How do they make sure that azobenzene is not adsorbed on the surface of the MOF films? This would also causes changes in the IR and UV/Vis signatures and enable photoswitching.

Response: We deeply appreciate the question raised by the reviewer. There are two main experimental results, which prove that azobenzene is incorporated within the host pores: When comparing the diffraction pattern of simulated *cis*- and *trans*-azobenzene with the experimental Zn-MOF film diffraction pattern, there is no presence of crystallized azobenzene on the film surface (see graph below). We further show in Figure S13 the shrinkage of the Zn-MOF film structure upon azobenzene uptake and the extent of it has been earlier discussed to depend on the functionalization and thus on the structural flexibility. Both experimental findings unequivocally demonstrate that the azobenzene molecule is infiltrated. This is further corroborated by the significant lower CO₂ uptake by the films because of pore blockage due to azobenzene, as it was quantified by QCM (see Figure S14).

Comment: In Figure 6 f it becomes apparent that the CO₂ uptake decreases with every irradiation step. The authors do not comment on this and do not explain the reason for this, which should be provided.

Response: We kindly thank the Reviewer for the remark and as this point has been risen by another Reviewer, we have included two additional graphs in the Supplementary Information to explain our findings better (Figure S12 c-d): The decline in sorption capacity is mainly related to the decrease in the photo-switching fraction of *trans-to-cis* azobenzene. In the case of Zn₂Me-BDC₂DABCO, when comparing the first and second photo-switch, IR results show only a decrease by 2% (Figure S12 c). In contrast, for Zn₂MeO-BDC₂DABCO, this decrease is significantly higher with 11% (Figure S12 d). This decrease is attributed to the functional groups present in the Zn-MOF structure, as this behaviour was not found in the non-functionalized system reported previously by some of the present authors (Klokic, S. et al. *Chemical science* 13, 11869–11877, (2022)).

To clarify this point raised by the Reviewer, we have included the following graphs in Figure S12 c-d:

Figure S12 (...) (c) Zn₂Me-BDC₂DABCO shows 21% of azobenzene photo-switching after the first cycle and 19% after the second. (d) Zn₂MeO-BDC₂DABCO shows 22% of azobenzene photo-switching after

the first cycle and 9% after the second. This decrease is attributed to the functional groups present in the Zn-MOF structure, as this behaviour is not found in the non-functionalized system.²

And we have included an explanation in the main manuscript in *lines 543 – 544*:

The decrease in sorption capacity is related to the lowering of the trans-to-cis isomerization fraction after the first photo-switch (Figure S12, c-d).

Reviewer #4 (Remarks to the Author): In this work, the authors combine three operando methods to investigate the CO₂ uptake and the structural changes it engenders in a library of zinc-based metal-organic framework (MOF) films which were grown hetero-epitaxially on Cu₂BDC₂-on-Cu(OH)₂ substructures, building on earlier literature reports for the synthesis of these MOF films. The novelty of this work lies in the expert combination of grazing incidence wide-angle x-ray scattering, Fourier-transformed infrared spectroscopy, and quartz-crystal microbalance with dissipation monitoring to inform on the structure and orientation of the MOF films, their characteristic infrared fingerprints, and the CO₂ uptake triggered by either changes in temperature or light, using the conformational change of azobenzene adsorbed in the MOF films for the latter. By considering five different linkers in the Zn₂L₂DABCO MOF system, the authors could modify the degree of MOF orientation on the substrate, the structural flexibility of the MOF, and the total CO₂ uptake. Overall, this interesting work demonstrates the strength of combining these three techniques, which may find promise beyond the promising materials studied herein. However, in its current version, the manuscript lacks mechanistic insight into what drives the preferred orientation and the structural changes upon CO₂ adsorption (vide infra), which seems necessary for publication in this high-profile journal.

Comment: From Figures 3 and S1d, it seems all MOF crystals form square rectangular prisms, the largest dimension of which is called the length L, and the smallest dimension of which is called the width W. However, the third dimension can be equal to either the length or the width; it is unclear throughout the text when one or the other possibility occurs. Furthermore, Figure 2 reports the fraction of crystals with oriented growth which, if I understand correctly, correspond to crystals whose long dimension (the length) falls perpendicular to the plane of the MOF/substrate interface (= the MOF film thickness). However, the SEM micrographs in Figure 3 show a strong variation in both the crystal dimensions and the orientation, even within a given MOF system. As a result, only providing the fraction of crystals that show oriented growth seems insufficient, especially when trying to understand the resulting flexibility of the material upon CO₂ adsorption. Hence, it would tremendously help the discussion if the authors could provide, for the four MOF films that show growth, a distribution of the crystal cell dimensions and the angle between the shortest dimension of the crystal and this interfacial plane, as the latter directly relates to the oriented growth.

Response: We kindly thank the Reviewer for raising this question and we have followed the suggestions by evaluating the in-plane alignment of the Zn-MOF films with respect to the lower Cu₂BDC₂ structure. To this aim, we have performed azimuthal angle dependence measurements (φ -scans) employing synchrotron radiation with results provided in Figure S4, where the intensities related to the (100), (001) and (110) reflections of the Zn-MOFs were recorded in the in-plane direction and their alignment was correlated to the Cu₂BDC₂ sublayer (0° - 270°). The Cu₂BDC₂ structure is grown on the sacrificial Cu(OH)₂ system, and its (100)_{Cu₂BDC₂} plane typically shows a strong alignment over the entire substrate (see references 35 and 49 in main manuscript), which was also observed for the herein investigated Zn-MOF films.

Based on these results, both the Zn₂MeO-BDC₂DABCO and the Zn₂Me-BDC₂DABCO systems align with their *a*-axis being parallel to the *a*-axis of the Cu₂BDC₂ structure (as (100)_{Zn₂L₂DABCO} and (100)_{Cu₂BDC₂} are coinciding in φ -scan), and orthogonally to this, their *c*-axes match the Cu₂BDC₂ *b*-axis (as (001)_{Zn₂L₂DABCO} is 90° shifted with respect to (100)_{Cu₂BDC₂} in φ -scan). The resulting lattice alignment is schematically depicted in the Supplementary Information in Figure S4 (also shown below). The Zn₂BDC₂DABCO structure shows this alignment, however it comprises also a 90° rotated lattice matching in which the *a*-axis matches the *b*-axis of the Cu₂BDC₂ system and orthogonally to it the *c*-axis aligns with the *a*-axis of the sublayer. This rotation in alignment can explain the observed differences in structural flexibility: The *c*-axis of the Zn-MOF bears no covalently interlinked

components since only the pillaring DABCO molecule bridges between the zinc paddlewheel units. Similarly, the *b*-axis of Cu₂BDC₂ bears no covalent interlinking amongst the Cu₂BDC₂ sheets and is often referred as to provide more flexibility to the MOF structure (*Falcaro et al., Nature materials, 16(3), 342-348 (2017)*). If both rather flexible axes are coinciding parallel to each other, this direction of the lattice allows considerably higher flexibility, which can also explain the success of this alignment despite the rather high mismatch ratio (see Table S2). In contrast, if the *a*-axis of the Zn-MOF, which comprises covalently interlinked terephthalate units with zinc-paddlewheel moieties, aligns onto the *c*-axis of Cu₂BDC₂, the flexibility of the entire film system is compromised. This alignment allows both the Zn₂MeO-BDC₂DABCO and the Zn₂Me-BDC₂DABCO structures to have a higher structural flexibility, as it is seen also in the CO₂ sorption measurements. This lattice alignment results in about 75% for Zn₂MeO-BDC₂DABCO and 66% for Zn₂Me-BDC₂DABCO of crystallites orient with one of their long axes being perpendicular to the substrate, with a degree of out-of-plane alignment of 22° and 34°, respectively. The width of the crystallite (shorter axis) aligns parallel to the substrate as does the third axis which in fact equals in length to the one orienting orthogonal to it. This is schematically outlined in Figure S1, where the three relevant axes (length (*ℓ*), length (*ℓ*) and width (*w*)) are indicated. As this aspect was agreeably not clear in the main manuscript, we have now indicated in Figure 3c that the third axis equals the length of the Zn-MOF crystal (*ℓ*), and have added in lines 324 – 327:

For the isostructural and heteroepitaxial Zn-MOF films, orientation analysis showed that 80% of the Zn₂MeO-BDC₂DABCO crystallites and 72% for Zn₂Me-BDC₂DABCO orient in out-of-plane direction with their long axis being perpendicular to the substrate (see ESI chapter 6 and Figure S5).

On a note, the amino-functionalized Zn-MOF structure shows only two-dimensional growth as already described in more detail in Figure S3, where azimuthal angle dependence data confirmed a similar lattice alignment as for the three-dimensional Zn₂L₂DABCO systems (see also main manuscript, lines 313 – 323).

We have included a more detailed discussion on the lattice alignment in the Supplementary Information in chapter 6, and have modified the sentences (294 – 312) in the main manuscript accordingly:

*Azimuthal angle dependence measurements confirmed a crystal alignment in which the *a*-axis for all three Zn₂L₂DABCO systems matches the *a*-axis of the Cu₂BDC₂-on-Cu(OH)₂ substructure, and orthogonally to that the *c*-axis aligns with the *b*-axis, respectively (Figure S4, d-e). For Zn₂MeO-BDC₂DABCO and Zn₂Me-BDC₂DABCO this alignment is supported by a low lattice mismatch in the *a*-axis direction (3.3% and 2.3%, see Table S2).⁴⁹ Considering the lattice parameters evaluated from the GIWAXS pattern with the results provided in Table S1, the lattice mismatch in *c*-axis direction of the upper Zn-MOF structures reaches about 19%. Interestingly, the Zn₂BDC₂DABCO structure comprises a second alignment by which the *a*- and *c*-axes are rotated in the in-plane direction by 90° (Figure S4, d). As both the methyl and methoxy-functionalized Zn-MOF structure lack this flip in alignment, the absence of this orientation is attributed to the bulkier functional groups and thus larger lattices as evidenced by an increase in the *a* and *b*-axis parameters (see Table S2), which could be directing the alignment of the upper Zn₂L₂DABCO structure. Because of this lattice orientation for both Zn₂MeO-BDC₂DABCO and Zn₂Me-BDC₂DABCO with respect to the lower Cu₂BDC₂ structure, both systems result in being more flexible compared to the Zn₂BDC₂DABCO system (see ESI, chapter 6 for more details). This result confirms that although the three Zn-MOF structures are grown isostructural, the functionalization of the Zn-MOF structures enforces a flip in epitaxial alignment, whilst an increase of the lattice constants causes a growing lattice mismatch with respect to the smaller substructure.*

And in lines 487 – 490, we have included:

This difference in behaviour is attributed to an increase in structural flexibility owed to the crystal lattice alignment that is also related to the linker functionalities (*vide supra*, Figure 3e),^{51,65} which further increase the amount of adsorbed CO₂ stems due to interactions arising between the adsorbate and the functionalized Zn-MOF structure.

Figure S1 Out-of-plane, in-plane and radial integration of the GIWAXS pattern for (a) Zn₂BDC₂DABCO (b) Zn₂MeO-BDC₂DABCO and (c) Zn₂Me-BDC₂DABCO. In-plane integrated GIWAXS pattern of the (100), (001), (110) or (101) reflections for Zn₂L₂DABCO and of the (001) reflection for the Cu₂BDC₂ substructure as a function of the rotation angle ϕ (azimuthal angle). The film rotation was conducted between 0 – 270°, for (d) Zn₂BDC₂DABCO, (e) Zn₂MeO-BDC₂DABCO, (f) Zn₂Me-BDC₂DABCO. Based on these GIWAXS data, the orientation of the films was deduced, and the preferential alignment of the upper Zn₂L₂DABCO structure with respect to the underlying Cu₂BDC₂-on-Cu(OH)₂ systems is displayed in the schematic in (g).

Comment: Figure 3f is confusing. First, the Cu₂BDC₂ substructure shows BDC linkers surrounding the Cu₂ paddlewheel units in all three directions despite being a 2D MOF. The substructure shown in this schematic is Cu₂BDC₃ instead, with an extra BDC linker binding to the axial Cu₂ paddlewheel positions. If this were correct, how would this binding take place? Second, the b direction in Cu₂BDC₂ is about half that of the Zn-MOFs (see Table S1), so every other Cu₂BDC₂ layer should remain disconnected from the Zn-MOF. Finally, the DABCO ligand should connect to the axial positions in the MOF; the four paddlewheel positions being occupied by BDC linkers.

Response: We thank the Reviewer for pointing out this error in the schematics provided in Figure 3f. We have redrawn the structural alignment of the upper Zn-MOF with respect to the Cu_2BDC_2 structure according also to the results of the azimuthal angle dependence measurements, and have also provided a front and side-view in the Supplementary Information in Figure S4 g, as provided below:

Figure 3. SEM micrographs and GIWAXS pattern of the Zn-MOF films. (a) SEM micrographs of $\text{Zn}_2\text{BDC}_2\text{DABCO}$, (b) $\text{Zn}_2\text{Me-BDC}_2\text{DABCO}$, (c) $\text{Zn}_2\text{MeO-BDC}_2\text{DABCO}$, (d) $\text{Zn}_2(\text{NH}_2)_2\text{-BDC}_2\text{DABCO}$. The length (l) and width (w) of the crystallites are indicated by arrows (c). (e) GIWAXS pattern evaluated for the out-of-plane direction for $\text{Zn}_2(\text{NH}_2)_2\text{-BDC}_2$, $\text{Zn}_2\text{MeO-BDC}_2\text{DABCO}$, $\text{Zn}_2\text{Me-BDC}_2\text{DABCO}$ and $\text{Zn}_2\text{BDC}_2\text{DABCO}$. The reflections corresponding to the Zn-MOF lattice are indicated by the areas highlighted in grey. The asterisk denotes reflections related to the $\text{Cu}_2\text{BDC}_2\text{-on-Cu(OH)}_2$ substructure.³⁵ (f) Schematics for the heteroepitaxial growth of the $\text{Zn}_2\text{L}_2\text{DABCO}$ structures along the in-plane and the out-of-plane directions ($L = \text{BDC}$, MeO-BDC and Me-BDC , a side-view is provided in Figure S4, g).

Figure S4 Based on these GIWAXS data, the orientation of the films was deduced, and the preferential alignment of the upper Zn_2L_2DABCO structure with respect to the underlying Cu_2BDC_2 -on- $Cu(OH)_2$ systems is displayed in the schematic in (g).

Comment: Can the authors relate the observed frequency shifts upon CO_2 or azobenzene adsorption to changes in cell parameters and, hence, flexibility? Do any events induce a partial transition from the LP to the CP phase? Can they quantify the statement on page 23 reading “This difference in behaviour is attributed to an increase in structural flexibility owed to introduced linker functionalities”?

Response: We thank the Reviewer for the questions and comments, and to answer those, we have followed the suggestion and compiled a comparison between IR and GIWAXS results. In the table below, the frequency shifts denote the difference between the non-infiltrated and azobenzene infiltrated Zn-MOF structures with the respective spectra provided in Graph 1 (A = BDC, B = Me-BDC, C = MeO-BDC), the changes in the d-spacing were extracted from the respective out-of-plane GIWAXS pattern (see also Figure S13), while the QCM-D data are included to support our line of argumentation.

When considering the frequency shift $\Delta\nu_{N-C-H}$ (related to a vibrational mode of DABCO, see main manuscript) versus the $\Delta\nu_{sym}$ (symmetric stretching of carboxy group, see main manuscript), the two modes give at first contradicting tendencies and a clear statement on flexibility trends based only on the vibrational modes is difficult. This is because these shifts are related to how the vibrational frequency is influenced for the respective bonds during elongation/contraction of the moieties in order to accommodate the azobenzene within the Zn-MOF pores. Based on this line of argumentation, one could draw an erroneous conclusion that the methyl-functionalized structure is more flexible than for example the methoxy one.

To properly relate these trends, it must be taken into account that the lattice constants increase with the bulkiness of the functional groups (in the order BDC < Me-BDC < MeO-BDC, see Table S1); and the comparably small frequency shifts for the methoxy functionalized structure are explained by the already spacious pores. Similarly, for the non-functionalized structures, the pores are simply too small and require an elongation. GIWAXS results however show a very strong shift in d-spacing after azobenzene infiltration indicating a contraction of the structure in the case of the methoxy functionalized structure (-2% for (100) and -1% for (001); -0.1% in IR) and a moderate for methyl (-0.7% in (100); -0.4 % in IR), whilst the non-functionalized structure elongates in the (100) direction thus straining the structure somehow in order to accommodate the *trans*-azobenzene molecule (+1.3% for (100); +0.1% in IR). Hence, in the case of azobenzene uptake, these percentages given for the infrared frequency shifts and the changes in cell parameters can serve to quantify the increasing flexibility in the order of MeO-BDC (-3%) > Me-BDC (-0.7%) > BDC (+1.3%). It must be noted that for Zn_2BDC_2DABCO , an occupation of one azobenzene molecule per Zn-MOF pore causes the elongation of the structure, while for the Me-BDC (0.7) and particularly MeO-BDC (0.2) linkers, a much lower loading level already causes a contraction of the structure (see Figure S13). This contraction (LP to NP) is presumed to occur fully for the entire Zn-MOF structure as no residual reflection at the initial positions are found.

L	IR-spectromicroscopy		GIWAXS		QCM-D
	$\Delta\nu_{sym} (cm^{-1})$	$\Delta\nu_{N-C-H} (cm^{-1})$	$\Delta d(100) (\text{Å})$	$\Delta d(001) (\text{Å})$	$\Delta m(CO_2) (\mu g/cm^2)$
BDC	+1.2	3.2	+0.13	---	0.09
Me-BDC	-5.1	2.5	-0.08	<0.1	0.42
MeO-BDC	-1.9	---	-0.22	0.1	0.50

^a (-) denotes a red-shift, (+) a blue-shift.

However, in the case of only CO₂ sorption, a quantification of the differences in flexibility based only on a similar direct comparison is troublesome as the structural flexibility is also influenced by the interactions between the functionalities and adsorbates: In the case of only CO₂, MeO-BDC causes an increase of about 2% for the *a* and *b* cell axes (with respect to BDC, see Table S1), this results in about 74% higher CO₂ mass uptake (Me-BDC with 0.9% about 69%).

Hence, only a combination of the experimentally observed parameters (e.g., frequency shift, reduction of d-spacing, losses in mass, loading level) allows to assess the extent of structural flexibility, which is in line with the argumentation presented in our manuscript. We have attempted to quantify the structural flexibility by considering the reduction in CO₂ uptake prior and after azobenzene infiltration as evaluated from QCM-D measurements. This can be done, as after azobenzene infiltration and although different loading levels are present, the three film structures adopt similar cell parameters as most of the reflections are overlapping (see Graph 2), which is the reason for the similar CO₂ sorption behaviour (see Figure S14). For MeO-BDC, a total cell parameter shrinkage of -3% resulted in a decrease of 0.5 μg/cm² in CO₂ uptake, whereas Me-BDC with only -0.7% shrinkage reduce the CO₂ uptake by 0.42 μg/cm², while the elongation of the pores for BDC results in a small reduction of 0.09 μg/cm² (see Table S3). We have also attempted to quantify the differences in structural flexibility in our low-temperature study, as discussed in detail in the main manuscript (see Table 1), based on which we still rule MeO-BDC as the most flexible structure as a 36% higher shift of the ν_{CO₂-ad} vibrational mode was observed.

Regarding the question if any events can cause a partial LP to NP transition, we are inclined to agree that most even at these low CO₂ pressures the shifts of the reflections of our GIWAXS results indicate such a partial transition (as seen in the main manuscript, Figure 5).

As we work with multilayer film structures anchored to the substrate, we usually observe especially for the (100) and (001) reflections a peak overlap, sometimes already partially present in the non-infiltrated structure. This is indicative that after azobenzene infiltration one does not observe a full LP to NP transition, and the non-infiltrated structures are a mixture of both as well (see Figure S13).

Graph 1: Comparison IR spectra for the Zn-MOF films prior and after azobenzene infiltration with A = BDC, B = Me-BDC and C = MeO-BDC.

Graph 2: Comparison of diffraction pattern for the azobenzene infiltrated Zn-MOF films.

Comment: Can the authors report on the location and distribution of the adsorbed CO₂ molecules both within the MOF film and within a MOF unit cell? A similar question holds for azobenzene since the loading level indicates only a fraction of the unit cell is occupied. Do azobenzene-loaded unit cells conglomerate? Where are the 13% of the azobenzene molecules that isomerise located?

Response: We kindly thank the Reviewer for the inquiries, and we would like to elaborate, as why most of the questions are troublesome to assess purely from an experimental approach.

A possibility in allocating the guest location and amount within a MOF framework may be inferred from differences in the X-ray diffraction (XRD) reflection intensities, which can be extracted through Rietveld refinements. For powdered samples, a forward structure solution can be employed to identify regions within the MOF structure likely to contain atoms. By assigning the correct structure factor phases to the structure factor amplitudes, a surface envelope can be generated *via* Fourier transformation. These surfaces are divided in regions with high and low electron density, which can aid for the structural identification of MOFs and potential guest molecules using solely powder diffraction data. (Brenner, S.; McCusker, L. B.; Baerlocher, C. J. *Appl. Crystallogr.*, 30, 1167 (1997)). By comparing the structure envelope of a MOF powder to its ideal structure (*e.g.*, from single-crystal diffraction), differences between the two envelopes can provide insights into the location, loading, and distribution of guest molecules, particularly if the difference envelope density is non-zero (Yakovenko, A. A.; Wei, Z.; Wriedt, M.; Li, J.-R.; Halder, G. J.; Zhou, H.-C.; *Crystal Growth & Design*, 14 (11), 5397-5407 (2014)).

However, such an analysis is challenging for powder MOFs due to the disordered nature of guest molecules within the pores, leading to peak overlaps and weak diffraction intensity differences. Obtaining clear results can be difficult even with synchrotron powder diffraction data (Yakovenko, A. A.; Wei, Z.; Wriedt, M.; Li, J.-R.; Halder, G. J.; Zhou, H.-C.; *Crystal Growth & Design*, 14 (11), 5397-5407 (2014)). For MOF films, such an analysis is even more complex, as peak intensities may be influenced by peak broadening, strain effects, Lorentz correction, and other factors, making the interpretation of diffraction data challenging. In the case of the Zn-MOF films studied herein, the

incorporation of azobenzene causes the formation of a nanoporous (NP) structure, results in a splitting of the reflection (see Figure S13), which complicates further precise analysis due to peak convolution.

At present, and especially in the case of the Zn-MOF films, we are unable to provide experimental evidence on the precise location of the azobenzene molecules or whether unit cells tend to conglomerate.

Comment: *Did the authors carry out GIWAXS experiments after the CO₂ has been evacuated from the MOF in Figure 5? This is necessary to distinguish between reversible and irreversible changes in the MOF.*

Response: This aspect was agreeably not very clearly stated in the present manuscript, and we would like to thank the Reviewer for the inquiry. We did perform these experiments with the traces given in black in the GIWAXS pattern presented in Figure 5. The measurements were conducted by acquiring the diffraction pattern in the beginning to CO₂ exposure (CO₂ ON (start)), by the end (CO₂ ON (end)) as well as after subsequent purging by N₂ (CO₂ OFF).

For a more precise clarification, we have modified the following sentence (*lines 467 – 468*):

Measurements were taken in the beginning (CO₂ start), after sufficient CO₂ load (CO₂ end) and after subsequent purging with N₂ (CO₂ OFF) considering the out-of-plane (OP) and the in-plane (IP) direction.

Comment: *What causes the distinct drop in CO₂ uptake both during CO₂ loading and upon switching in Figure 6e,f? Did the authors check the cyclability of temperature-induced CO₂ adsorption?*

Response: We kindly thank the Reviewer for pointing out this response behaviour, which can be observed even on the bare QCM-D substrate (see zoom-in of Figure S6a, black trace). The transient response of the signal for the bare sensor is interpreted as a change in pressure conditions when switching from N₂ to CO₂, which influences the QCM-D sensor's oscillation. The sensor accelerates when CO₂ is removed and decelerates when CO₂ is added, which is related not only to the mass on the surface but also to the density of the gas phase and therefore a pressure change.

As closely related questions to this topic have been risen by other Reviewers, we have included the explanation to this behaviour in the Supplementary Information in Figure S6, legend:

The signal jumps likely reflect pressure changes when switching between N₂ (CO₂ ON) and CO₂ (CO₂ OFF), affecting sensor oscillation.

In the case of Zn-MOF films, the comparably much stronger increase of the response signal is probably also related to the swelling of the layer due to CO₂ adsorption followed by its stabilization of this new configuration. This is typically observed in QCM-D measurements, as exemplarily given in reference S12. We have included this explanation in chapter S8 in the Supplementary Information:

In the case of the Zn-MOF films the type of response shown in Figure S6 indicates that the only the MOF layer is highly sensitive towards CO₂. This is evidenced as a rapid increase in the response occurs due to the adsorption of the analyte caused by swelling of the Zn-MOF film and followed by a drop in response until the stabilization of the CO₂ adsorbed layer.¹²

And we have referenced this aspect also in the main manuscript in lines 387 – 388:

Results showed an immediate response of the Zn-MOF films towards CO₂ with strong and rapid gas adsorption, much in contrast to the responses obtained for the bare crystal or Cu₂BDC₂ substructure (see Supplementary Information chapter 8 and Figure S6, a).

For the temperature-induced CO₂ sorption measurements, infrared spectra were acquired at five spatially different positions on the sample for every temperature during CO₂ ad- and desorption (see line 198 – 199, Methods). This protocol provides a more representative average on the response signal as different areas are probed. In addition, we performed this protocol at 295 K as triplicate measurement, whilst at lower temperature duplicate measurements were performed (240 K, 200 K). The latter is because the CO₂ uptake and especially its release were extremely slowed at low temperatures, and as these measurements were performed at a synchrotron with limited temporal access to the instruments, the number of repetitions was reduced to two (with 5 points per sample) to still acquire sufficient statistics.

To highlight this point that CO₂ sorption cyclability has been performed, we have added this information in line 207:

This procedure was repeated at 295 as triplicates, at 240 K and 200 K for time reasons as duplicates.

Comment: The authors denote the absence of a band around 1690 cm⁻¹ to the full conversion of -COOH groups to -COO- groups. However, for the Cu₂BDC₂ substrate, shouldn't these -COOH groups still be present at the surface where the Zn-MOF will grow in a subsequent step?

Response: We kindly thank the Reviewer for highlighting this issue and we do observe such residual -COOH groups remaining present on the film's surface. For example, the Cu₂BDC₂ infrared spectrum in Figure S2 does indeed show a very weak mode at this wavenumber. Moreover, the Zn-MOF structures with the ligands BDC, Me-BDC and MeO-BDC also show such a feature as indicated by an arrow in the graph below. Our rather unfortunate choice in words implied that the absence of a *strong* vibrational bands at 1690 cm⁻¹ can be interpreted as a complete lack of such.

As we realize that this choice of words is confusing, we have reformulated this sentence (supplementary information, legend of Figure S2) to:

The weak mode at 1690 cm^{-1} indicates that the $-\text{COOH}$ group is converted to $-\text{COO}^-$ for the functionalized BDC ligand,⁵ thus its coordination to the zinc-metal nodes was successful.

Comment: When discussing the fabrication of the pristine Zn-MOF films, no source of zinc ions is present when immersed in the linker-based solutions (lines 152-157). How does the MOF growth then take place?

Response: We thank the Reviewer for the remark and would like to note that we use an ethanolic zinc acetate solution as stated in line 151 into which the $\text{Cu}_2\text{BDC}_2\text{-on-Cu}(\text{OH})_2$ film is immersed. This step is done to covalently bind the zinc nodes (see line 150 – 151). After this step, the films are removed from this solution and immersed into the linker solutions as stated in lines 153 and following.

To make this experimental step more concise, we have included in lines 150 – 152:

In the first step, the conversion of $\text{Cu}_2\text{BDC}_2\text{-on-Cu}(\text{OH})_2$ in a methanolic zinc acetate solution (2.2 mg, 10 mL of EtOH) was done to covalently bind Zn^{2+} that acts as the metal node for the Zn-MOF growth.³³

Comment: Why was the nonfunctionalised Zn₂BDC₂DABCO not considered for azobenzene loading?

Response: We thank the Reviewer for the inquiry; however, we must note that we did consider this structure for azobenzene loading with results shown in Figure S14 (c) and Figure S15. Results related to CO_2 sorption prior and after azobenzene infiltration are further summarized in Table S3. We did realize that the respective GIWAXS pattern was missing, and we have included one now in Figure S13 (c).

Figure S2 Infiltration of azobenzene (AB) into the Zn-MOF film structures. (...) (c) The Zn_2BDC_2DABCO structure shows a shift by $\Delta q_{(100)} = 0.07 \text{ nm}^{-1}$ towards larger d -spacing upon incorporation of AB molecules indicating that the structure expands.

Comment: Lines 150-151 mention “methanolic” and “EtOH” simultaneously.

Response: We kindly thank the Reviewer for pointing out this error in the Methods section and we have corrected the issue accordingly:

In the first step, the conversion of Cu_2BDC_2 -on- $Cu(OH)_2$ in a methanolic zinc acetate solution (2.2 mg, 10 mL of MeOH) was done to covalently bind Zn^{2+} that acts as the metal node for the Zn-MOF growth.³³

Comment: Is there a reason that a Delta symbol precedes the frequencies in Figure 4a, b, and h, since they show the actual frequencies and no frequency shifts?

Response: The Delta symbols indeed refer to the discussion of the red- and blue-shifts, where a comparison between the three Zn-MOF films is drawn in the main text in lines 357 – 367. We thank the Reviewer for the inquiry as the symbols are not mentioned in the legend in Figure 4, which is now clarified in lines 382 – 383:

The $\Delta\nu$ symbols refer to shifts between the functionalized Zn-MOF films (see main text).

Responses to Reviewer #1: (Remarks to the Author): We are happy with the response from the authors.
Comment: One minor issue is that the SEM in Figure 3D is significantly charged, which makes it difficult to clearly support the claim in the context.

Response: We kindly thank the Reviewer for highlighting this issue of overcharging in Figure 3d. To tackle this issue and to support our claim in the main text, we have remeasured this sample carefully by means of SEM and have included now a new micrograph for the structure $Zn_2(NH_2)_2-BDC_2DABCO$ in Figure 3d, as shown below:

Figure 3. SEM micrographs and GIWAXS pattern of the Zn-MOF films. (a) SEM micrographs of Zn_2BDC_2DABCO , (b) $Zn_2Me-BDC_2DABCO$, (c) $Zn_2MeO-BDC_2DABCO$, (d) $Zn_2(NH_2)_2-BDC_2DABCO$. The length (l) and width (w) of the crystallites are indicated by arrows in (c). (e) GIWAXS pattern evaluated for the out-of-plane direction for $Zn_2(NH_2)_2-BDC_2DABCO$, $Zn_2MeO-BDC_2DABCO$, $Zn_2Me-BDC_2DABCO$ and Zn_2BDC_2DABCO . The reflections (100), (001), (110) and (200) corresponding to the Zn-MOF lattice are indicated by the areas highlighted in grey. The asterisks denote reflections related to the $Cu_2BDC_2-on-Cu(OH)_2$ substructure.³⁵ (f) Schematics for the heteroepitaxial growth of the Zn_2L_2DABCO structures along the in-plane and

the out-of-plane directions ($L = \text{BDC}$, Me-BDC and MeO-BDC , a side-views are provided in Figure S4, g). The $(00l)$ and $(h00)$ planes align in-plane (orange highlighted areas), whilst the $(0k0)$ plane (green highlighted area) orients in the out-of-plane direction being parallel to the substrate. Source data are provided as a Source Data file.

We have further included in Figure S2, e-f, a zoom-in onto a different part of the sample to provide another representative shape of these crystallites, and onto the lower, needle alike shaped, Cu_2BDC_2 structure.

When comparing the shape of the crystal for $\text{Zn}_2(\text{NH}_2)_2\text{-BDC}_2\text{DABCO}$ with those of the other functionalized Zn-MOF structures shown in Figure 3, a-c, this system grows rather as a platelet-like system and not as a cuboid crystallite.

Reviewer #4 (Remarks to the Author): I thank the authors for their clarifications and revisions in the manuscript, which cover the initial reservations I had. As a result, I'm happy to suggest this manuscript for publication in Nature Communications.

Comment: As a final suggestion, I would encourage the authors to include in their final discussion not only the strength of combining the three reported techniques but also the main challenges that remain unresolved, for instance, how to probe the distribution of the adsorbed CO₂ molecules in the MOF film.

Response: We kindly thank the Reviewer for this excellent suggestion, which we have followed by including the following sentence in the final discussion:

Results showed that because of the adsorbed azobenzene and CO₂ molecules, interactions arise which are accredited to aid the system in adapting to the photo-response. Overall, these findings highlight that the stimuli-responsive behaviour of MOF films under CO₂ load can be readily explored by the experimental techniques proposed herein. Challenges remain especially when it comes to directly map the spatial distribution of CO₂ molecules within the films at the nanoscale. Addressing this limitation could stimulate the development of characterization techniques that are essential for advancing our understanding of transport and storage mechanisms in MOF films. Finally, the methodologies used in this study offer a general conceptual advance for the investigation of structural and molecular changes under operando conditions in stimuli-responsive films, which are applicable also to more complex systems such as mixed matrix membranes⁹ or assemblies in device configuration.⁴